# Exploration of machine learning methods for the classification of infrared limb spectra of polar stratospheric clouds

Rocco Sedona[1,4], Lars Hoffmann[1], Reinhold Spang[2], Gabriele Cavallaro[1], Sabine Griessbach[1], Michael Höpfner[3], Matthias Book[4], and Morris Riedel[4]

[1]Jülich Supercomputing Centre (JSC), Forschungszentrum Jülich, Jülich, Germany
[2]Institut für Energie- und Klimaforschung (IEK-7), Forschungszentrum Jülich, Jülich, Germany
[3]Institut für Meteorlogie und Klimaforschung, Karlsruher Institut für Technologie, Karlsruhe, Germany
[4]University of Iceland, Reykjavik, Iceland

**Correspondence:** Rocco Sedona (r.sedona@fz-juelich.de)

**Abstract.** Polar stratospheric clouds (PSC) play a key role in polar ozone depletion in the stratosphere. Improved observations and continuous monitoring of PSCs can help to validate and improve chemistry-climate models that are used to predict the evolution of the polar ozone hole. In this paper, we explore the potential of applying machine learning (ML) methods to classify PSC observations of infrared limb sounders. Two datasets have been considered in this study. The first dataset is a collection of infrared spectra captured in Northern Hemisphere winter 2006/2007 and Southern Hemisphere winter 2009 by the Michelson Interferometer for Passive Atmospheric Sounding (MIPAS) instrument onboard ESA's Envisat satellite. The second dataset is the cloud scenario database (CSDB) of simulated MIPAS spectra. We first performed an initial analysis to assess the basic characteristics of the CSDB and to decide which features to extract from it. Here, we focused on an approach using brightness temperature differences (BTDs). From both, the measured and the simulated infrared spectra, more than 10,000 BTD features have been generated. Next, we assessed the use of ML methods for the reduction of the dimensionality of this large feature space using principal component analysis (PCA) and kernel principal component analysis (KPCA) followed by a classification with the support vector machine (SVM). The random forest (RF) technique, which embeds the feature selection step, has also been used as classifier. All methods were found to be suitable to retrieve information on the composition of PSCs. Of these, RF seems to be the most promising method, being less prone to overfitting and producing results that agree well with established results based on conventional classification methods.

## 1   Introduction

Polar stratospheric clouds (PSC) typically form in the polar winter stratosphere between 15 and 30 km of altitude. PSCs can be observed only at high latitudes, as they exist only at very low temperatures ($T < 195$ K) found in the polar vortices. PSC are known to play an important role in ozone depletion caused by heterogeneous reactions under cold conditions, while denitrification of the stratosphere extends the ozone destruction cycles into springtime, as the absence of $NO_y$ limits the deactivation process of the reactive ozone destroying substances (Solomon, 1999; Salawitch et al., 1993). The presence of man-made chlorofluorocarbons (CFCs) in the stratosphere, which have been used for example in industrial compounds present in refrigerants,

solvents, blowing agents for plastic foam affects ozone depletion. CFCs are inert compounds in the troposphere, but get transformed under stratospheric conditions to the chlorine reservoir gases HCl and ClONO$_2$. PSC particles are involved in the release of chlorine from the reservoirs.

The main constituents of PSCs are three, i. e., nitric acid trihydrate (NAT), super-cooled ternary solutions (STS), and ice
(Lowe and MacKenzie, 2008). Michelson Interferometer for Passive Atmospheric Sounding (MIPAS) measurements have been used to study PSC processes (Arnone et al., 2012; Khosrawi et al., 2018; Tritscher et al., 2019). The infrared spectra acquired by MIPAS are rather sensitive to optically thin clouds due to the limb observation geometry. This is particularly interesting for NAT and STS PSCs, as ice PSCs are in general optically thicker than NAT and STS (Fromm et al., 2003). As ice clouds form at a lower temperature than NAT and STS, they are mainly present in the Antarctic, while their presence in
the Arctic (where the stratospheric temperature minimum in polar winter is higher) is only notable for extremely cold winter conditions (e. g., Campbell and Sassen, 2008; Pawson et al., 1995).

Besides using MIPAS measurements, classification has been carried out with different schemes based on the optical properties of PSCs by LIDAR measurements. A review of those methods is available in Achtert and Tesche (2014). Classification schemes are based on two features, namely the backscatter ratio and the depolarization ratio. As exposed in Biele et al. (2001),
particles with large backscatter ratio and depolarization are likely to be composed of ice (type II). Type I particles are characterized by a low backscatter ratio. The subtype Ia particles show a large depolarization and are composed of NAT, whereas subtype Ib particles have low depolarization and consist of STS. The threshold to classify the PSC types varies among different works such as Browell et al. (1990); Toon et al. (1990); Adriani (2004); Pitts et al. (2009, 2011). The nomenclature presented above is a simplification of real case scenarios, since PSCs can occur also with mixtures of particles with different composition
(Pitts et al., 2009). Other methods that are used to measure PSCs are in situ optical and non-optical measurements from balloon or aircraft as well as microwave observations (Buontempo et al., 2009; Molleker et al., 2014; Voigt, 2000; Lambert et al., 2012).

The use of machine learning (ML) algorithms increased dramatically during the last decade. ML can offer valuable tools to deal with a variety of problems. In this paper, we used ML methods for two different tasks. First, for the selection of informative features from the simulated MIPAS spectra. Second, to classify the MIPAS spectra depending on the composition
of the PSC. In this work we significantly extended the application of ML methods for the analysis of MIPAS PSC observations. Standard methods that exploit infrared limb observation to classify PSCs are based on "empirical" approaches. Given physical knowledge of the properties of the PSC, some features have been extracted from the spectra, as for example the ratio of the radiances between specific spectral windows. These approaches have been proven to be capable to detect and discriminate between different PSC classes (Spang et al., 2004; Höpfner et al., 2006).
The purpose of this study is to explore the use of ML methods to improve the PSC classification for infrared limb satellite measurements and to potentially gain more knowledge on the impact of the different PSC classes on the spectra. We compare results from the most advanced "emprical" method, the Bayesian classifier of Spang et al. (2016), with three "automatic" approaches. The first one relies on principal component analysis (PCA) and kernel principal component analysis (KPCA) for feature extraction, followed by classification with support vector machine (SVM). The second one is similar to the first, but
uses kernel principal component analysis (KPCA) for feature extraction instead of PCA. The third one is based on the random

forest (RF), a classifier that directly embeds a feature selection (Cortes and Vapnik, 1995; Breiman, 2001; Jolliffe and Cadima, 2016). A common problem of ML is the lack of annotated data. To overcome this limitation, we used a synthetic dataset for training and testing, the cloud scenario database (CSDB), especially developed for MIPAS cloud and PSC analyses (Spang et al., 2012). As a "ground truth" for PSC classification is largely missing, we evaluate the ML results by comparing them with results from existing methods and show that they are consistent with established scientific knowledge.

In Sect. 2, we introduce the MIPAS and synthetic CSDB data sets. A brief description of the ML methods used for feature reduction and classification is provided in Sect. 3. In Sect. 4, we compare results of PCA+SVM, KPCA+SVM, and RF for feature selection and classification. We present three case studies and statistical analyses for the 2006/2007 Arctic and 2009 Antarctic winter season. The final discussion and conclusions are given in Sect. 5.

## 2   Data

### 2.1   MIPAS

The MIPAS instrument (Fischer et al., 2008) was an infrared limb emission spectrometer onboard ESA's Envisat satellite to study the thermal emission of the Earth's atmosphere constituents. Envisat operated from July 2002 to April 2012 in a polar low Earth orbit with a repeat cycle of 35 days. MIPAS measured up to 87°S and 89°N latitude and therefore provided nearly global coverage at day- and nighttime. The number of orbits of the satellite per day was equal to 14.3, resulting in a total of about 1000 limb scans per day.

The wavelength range covered by the MIPAS interferometer was about 4 to 15 μm. From the beginning of the mission to spring 2004, the instrument operated in the full resolution (FR) mode ($0.025$ cm$^{-1}$ spectral sampling). Later on, this has to be changed to the optimized resolution (OR) mode ($0.0625$ cm$^{-1}$) due to a technical problem of the interferometer (Raspollini et al., 2006, 2013). The FR measurements were taken with a constant 3 km vertical and 550 km horizontal spacing, while for the OR measurements the vertical sampling depended on altitude, varying from 1.5 to 4.5 km, and a horizontal spacing of 420 km was achieved. The altitude range of the FR and OR measurements varied from 5-70 km at the poles and to 12-77 km at the equator.

For our analyses, we used MIPAS Level-1B data (version 7.11) acquired at $15-30$ km of altitude between May and September 2009 at $60-90$ °S and between November 2006 and February 2007 at $60-90$ °N. 2009 Southern Hemisphere winter presents a slightly higher than average PSC activity, especially for ice in June and August. 2006/2007 Northern Hemisphere winter is characterized by a large area covered by NAT, exception made for early January, and some ice is present in late December (this analysis was obtained from NASA Ozone Watch from their web site at https://ozonewatch.gsfc.nasa.gov ). The high-resolution MIPAS spectra were averaged to obtain 136 spectral windows of 1 cm$^{-1}$ width, because PSC particles are expected to typically cause only broader scale features. The $1\,\mathrm{cm}^{-1}$ window data used in this study comprise the 8 spectral regions reported in Table 1. In addition to these, five windows (W1-W5) larger than $1\,\mathrm{cm}^{-1}$ have been considered, as used in the study of Spang et al. (2016).

From the $1\,\mathrm{cm}^{-1}$ windows and the five additional larger windows, more than 10,000 brightness temperature differences (BTDs) have extracted using a two-step pre-processing. At first, the infrared spectra have been converted from radiance intensities to BTs. This approach is considered helpful, as variations in the signals are more linear in BT compared to radiances. Then, the BTDs have been computed by subtracting the BT of each window with respect to the remaining ones. The main

motivation for using BTDs rather than BTs for classification is to try to remove background signals from interfering instrument effects such as radiometric offsets.

Other wavelength ranges covered by MIPAS have been excluded here as they are mainly sensitive to the presence of trace gases. The interference of cloud and trace gas emissions makes it more difficult to analyze the effects of the PSC particles (Spang et al., 2016). As an example, Fig. 1 shows MIPAS spectra of PSC observations acquired in late August 2009 in Southern

Hemisphere polar winter conditions, with the spectral regions used for PSC detection and classification being highlighted.

## 2.2   Cloud Scenario Database

A synthetic data set consisting of simulated radiances for the MIPAS instrument provides the training and testing data for this study. The CSDB was generated by considering more than 70,000 different cloud scenarios (Spang et al., 2012). The CSDB spectra have been generated using the Karlsruhe Optimized and Precise Radiative Transfer Algorithm (KOPRA) model (Stiller

et al., 1998). Limb spectra have been simulated from 12 to 30 km tangent height, with 1 km vertical spacing. Cloud top heights have been varied between 12.5 and 28.5 km, with 0.5 km vertical spacing. The cloud vertical extent varies between 0.5, 1, 2, 4, and 8 km. The spectral features selected from the CSDB are the same as those for MIPAS (Sect. 2.1, Fig. 1).

As described in Spang et al. (2016), the CSDB was calculated with typical particle radii and volume densities of PSCs (Table 2). Five different PSC compositions have been considered: ice, NAT, STS with 2% $H_2SO_4$, 48% $HNO_3$ and 50% $H_2O$ (called

later on STS 1), STS with 25% $H_2SO_4$, 25% $HNO_3$ and 50% $H_2O$ (STS 2), STS with 48% $H_2SO_4$, 2% $HNO_3$ and 50% $H_2O$ (STS 3). This values are derived from the model by Carslaw et al. (1995) and span over all possible compositions. The CSDB does not give any representative frequency of real occurrences in the atmosphere. For this study, we decided to split the set of NAT spectra into two classes, large NAT (*radius* > 2µm) and small NAT (*radius* <= 2µm). This decision was taken to assess the capability of the classifiers to correctly separate between the two classes. It is well known that small NAT particles

(*radius* <= 2µm) produce a specific spectral signature at 820 cm$^{-1}$ (Spang and Remedios, 2003; Höpfner et al., 2006). Spectra for large NAT particles are more prone to overlap with those of ice and STS.

To prepare both, the real MIPAS and the CSDB data for PSC classification, we applied the cloud index (CI) method of Spang et al. (2004) with a threshold of 4.5 to filter out clear air spectra. In optimal conditions a *CI* < 6 detects clouds with extinction coefficients down to about $2 \times 10^{-5}$km$^{-1}$ in the mid-infrared (Sembhi et al., 2012). However, in the polar winter regions these

optimal conditions do not persist over an entire winter season. Hence, we selected a threshold of 4.5 that reliably discriminates clear air from cloudy air in the southern and northern hemisphere polar winter regions as it is sensitive to extinctions down to $5 \times 10^{-4}$km$^{-1}$ (Griessbach et al., 2020).

## 3 Methods

### 3.1 Conventional classification methods

Spang et al. (2016) provide an overview on various conventional methods used to classify Envisat MIPAS PSC observations. Furthermore, a Bayesian approach has been introduced in their study to combine the results of individual classification methods. This approach is used as benchmark for the new classifiers introduced in the present paper. The Bayesian classifier considers a total of 13 features, including correlations between the cloud index (CI) (Spang et al., 2004), the NAT index (NI) (Spang and Remedios, 2003; Höpfner et al., 2006), and another five additional BTDs. Each feature has been assigned individual probabilities $p_{i,j}$ in order to discriminate between the different PSC composition classes. The output of the Bayesian classifier is calculated according to $P_j = \prod_i p_{i,j} / \sum_j \left( \prod_i p_{i,j} \right)$, where the indices $i = 1, \ldots, 13$ and $j = 1, 2, 3$ refer to the individual feature and the PSC constituent, respectively. The normalized probabilities $P_j$ per PSC constituent are used for final classification applying the maximum a posteriori principle. The BC composition classes are the following: unknown, ice, NAT, STS_mix, ICE_NAT, NAT_STS, and ICE_STS. A stepwise decision criterion is applied to classify each spectrum. If the maximum of $P_j(with\,j = 1...3) > 50\%$, then the spectrum is assigned a single PSC composition label. If two $P_j$ values are between 40 and 50 %, then a mixed composition class, for example ICE_STS for $j = 1$ and $j = 3$, is attributed. If the classification results in P1, P2, or P3 $< 40\%$, then the spectrum is labeled as "unknown". Considering the Southern Hemisphere 2009 case, the NAT_STS mixed composition class is populated with more than 4000 spectra, while ICE_STS and ICE_NAT predictions are negligible (Fig. 2). The analysis of the complete MIPAS period (9 Southern Hemisphere and 10 Northern Hemisphere winters in Spang et al. (2018)) showed that ICE_STS and ICE_NAT classes are generally only in the sub percentage range and statistically not relevant. The Bayesian classifier requires a priori information and detailed expert knowledge on the selection of the features to be used as discriminators and in assigning the individual probabilities $p_{i,j}$ for classification. In this work, we aim at investigating automatic ML approaches instead of the manual or empirical methods applied for the Bayesian classifier. Nevertheless, being carefully designed and evaluated, the results of the Bayesian classifier are used for further reference and comparison in this study.

### 3.2 Feature extraction using PCA and KPCA

In a first step, we calculated BTDs from the $1\,\mathrm{cm}^{-1}$ downsampled radiances of the CSDB. Calculating the BTDs between the 142 spectral windows resulted in 10,011 BTDs for a total of 70,000 spectra. In a second step, in order to reduce the amount of data, we applied a variance threshold to exclude BTD features with relatively low variance ($\sigma^2 < 10\,\mathrm{K}^2$), as this indicates that the corresponding windows have rather similar information content. In order to further reduce the difficulties and complexity of the classification task, we decided to even further reduce the number of BTD features before training of the classifiers by means of feature extraction.

Feature selection methods are used for picking subsets of an entire set of features while keeping the information content as high as possible. The methods help to reduce the training time of the classifier and to reduce the risk of overfitting. Feature selection methods typically belong to three main families (Bolón-Canedo et al., 2016): (i) filter methods, where the importance

of the feature is derived from intrinsic characteristic of it, (ii) wrapper methods, where the features are selected by optimizing the performances of a classifier, and (iii) embedded methods, where classification and selection happen at the same time. Here, we used a more advanced approach to dimensionality reduction, which goes under the name of feature extraction. In this case, instead of simply selecting a subset of the original features, the set of features itself is transformed to another space where the selection takes place.

Principal component analysis (PCA) is among the most popular feature extraction methods (Jolliffe and Cadima, 2016). The main idea of the PCA is to reproject the data to a space where the features are ranked on the variance that they account for. At first a centering of the data through the subtraction of the mean is performed. Then, the covariance matrix is calculated and its eigenvectors and eigenvalues are computed. At this point, selecting the eigenvectors whose eigenvalues are largest, it is possible to pick the components on which most of the variance of the data lays. PCA already found applications in the analysis of atmospheric mid-infrared spectra, in particular for the compression of high-resolution spectra and for accelerating radiative transfer calculations (e. g., Huang and Antonelli, 2001; Dudhia et al., 2002; Fauvel et al., 2009; Estornell et al., 2013). PCA has been used in this study for two main purposes, dimensionality reduction and visualization of the data.

Kernel PCA (KPCA) is an extension of the PCA where the original data $\boldsymbol{x}$ are first transformed using a mapping function $\phi(\boldsymbol{x})$ to a higher dimensional feature space. The main advantage of using KPCA relies in the fact that it can capture non-linear patterns, which PCA, being a linear method, may fail to represent well. However the construction of the kernel matrix $K$ for mapping can be expensive in terms of memory. This latter problem undermines severely the possibility of using this algorithm for large datasets. At this point the kernel trick comes into play (Schölkopf et al., 1997). It helps to avoid the inconvenience of having to compute the covariance matrix in a large transformed space. Instead of translating each data point to the transformed feature space using the mapping function $\phi(\boldsymbol{x})$, the inner product can be calculated as $K(\mathbf{x}_i, \mathbf{x}_j) = \phi(\mathbf{x}_i)\phi(\mathbf{x}_j)$, resulting in a much less demanding computational task. Among the most common kernels there are the radial basis function (RBF) and the polynomial (Genton, 2002), which we also considered in this study.

### 3.3 Classification using Support Vector Machines and Random Forests

Supervised classification is a ML task in which the classes or "labels" of unknown samples are predicted by making use of a large data set of samples with already known labels. In order to do that, the classification algorithm has first to be trained, i. e., it has to learn a map from the input data to its target values. After a classifier is trained, one can give it as input an unlabeled set of data points with the aim of predicting the labels. The training of a classifier is usually a computationally demanding task. However, the classification of unknown samples using an already trained classifier is computationally cheap.

A large number of classifiers exists, based on rather different concepts. Bayesian classifiers follow a statistical approach. Support vector machines (SVMs) are based on geometrical properties. Random forests (RF) are based on the construction of multiple decision trees. Neural networks try to emulate the behaviour of the human brain by stacking a number of layers composed of artificial neurons (Zeiler and Fergus, 2014). According to the "no free lunch theorem", it is not possible to state safely which algorithm is expected to perform best for any problem (Wolpert, 1996). In this study, we selected two well established methods, RFs and SVMs, to test their performance.

Random Forest is an algorithm that learns a classification model by building a set of decision trees. A decision tree is composed of decision nodes, which lead to further branches and leaf nodes, which finally represent classification results. RFs are non-parametric models that do not assume any underlying distribution in the data (Breiman, 2001). RF builds a number of decision trees selecting a random subset of the original features for each tree. In this way the model becomes more robust against overfitting. The classification result of the RF model will be the label of the class that has been voted for by the majority of decision trees (Liu et al., 2012). An interesting characteristic of the RF classifier is that it can give by calculating the Gini index (Ceriani and Verme, 2012) also a measure of the feature importance. In this way, the RF classifier can also be exploited for performing feature selection.

The performance of a RF classification model depends on a number of hyperparameters, which must be defined before training: (i) The "number of estimators" or decision trees of the forest needs to be defined. (ii) A random subset of the features is selected by each decision tree to split a node. The dimension of the subset is controlled by the hyperparameter "maximum number of features". (iii) The "maximum depth", i. e., the maximum number of levels in each decision tree controls the complexity of the decision trees. In fact, the deeper a decision tree is, the more splits can take place in it. (iv) The "minimum number of samples before split" that has to be present in a node before it can be split also needs to be defined. (v) A node without further split, has to contain a "minimum number of samples per leaf" to exist. (vi) Finally, we have to decide whether to use "bootstrapping" or not. Bootstrapping is a method used to select a subset of the available data points, introducing further randomness to increase robustness (Probst et al., 2019).

SVMs became popular around the 90's (Cortes and Vapnik, 1995). The method is based on the idea of identifying hyperplanes, which best separate sets of data points into two classes. In particular, SVM aims at maximizing the margin, which is the distance between few points of the data, referred to as "support vectors", and the hyperplane that separates the two classes. The "soft margin" optimization technique takes into account the fact that misclassification can occur due to outliers. For that reason a tuning parameter $C$ is included in order to allow for the presence of misclassified samples during the optimization of the margin to a given extent. The choice of the parameter $C$ is a trade-off between minimizing the error on the training data and finding a hyperplane that may generalize better (Brereton and Lloyd, 2010).

SVM had been originally developed to find linear decision boundaries. However, the introduction of the kernel trick (cf., Sect. 3.2) enables the possibility for non-linear decision boundaries. Kernel functions, e. g., radial basis functions or polynomials, are mapping from the original space to a non-linearly transformed space, where the linear SVM is applied (Patle and Chouhan, 2013). In the case of a non-linear kernel, the parameter $\gamma$ is used to define how much a support vector has influence on deciding the class of a sample. A small value of $\gamma$ implies that this support vector also has impact on samples far in the feature space, a large value of $\gamma$ has an influence only on samples that are close in the feature space.

We recap in Fig. 3 the entire pipeline for training and prediction. The BTDs extracted from the CSDB dataset are given as input to the PCA or KPCA methods, and the extracted featured are fed to the SVM classifier for model training (PCA+SVM and KPCA+SVM). On the other hand, the RF classifier is given as input BTDs directly, without prior feature extraction. The input samples (BTDs) are annotated with a label as explained in Sect. 2.2. In prediction (Fig. 3b), the BTDs extracted from the MIPAS measurements are the input to the three methods PCA+SVM, KPCA+SVM and RF, where the output are

the predicted label for each sample. The RF classifier provides a feature importance measure as well. During prediction, the sample is assigned to one of the following classes representing the main constituent: ice, small NAT, large NAT, STS 1, STS 2,STS 3. Compared to the NAT class of the Bayesian classifier, in the proposed ML methods NAT particles are assigned to small and large NAT subclasses. The STS_mix class of the BC overlaps with STS 1, STS 2 and STS 3. There are not directly corresponding classes to the mixed composition ones of the BC. As discussed above in the text, only a few spectra are classified by the BC as ICE_STS or ICE_NAT. Samples belonging to the NAT_STS class of the BC, characterized by a non-negligible population, are labeled by the new ML classes mostly as STS 1 (Fig. 2).

## 4 Results

### 4.1 Feature extraction

In this study, we applied PCA and KPCA for feature extraction from a large set of BTDs. Both, PCA and KPCA are reprojecting the original BTD features to a new space, where the eigenvectors are ordered in such a way that they maximize variance contributions of the data. Figure 4a shows a matrix of the normalized variances of the individual BTDs considered here. The matrices in Fig. 4 are symmetric, thus the reader can either focus on the location (i.e. the indices of the BTs from which the BTD feature has been computed) of the maximum values in the upper or lower triangular part. A closer inspection shows that the largest variances originate from BTDs in the range from 820 to 840 cm$^{-1}$ (indicated as spectral region R1 in in Table 1) and 956 to 964 cm$^{-1}$ (R2). BTDs close to 790 cm$^{-1}$ (R1, BT index $\sim$ 10) also show high variance. Another region with high variances originates from BTDs between 820 to 840 cm$^{-1}$ (part of R1) and 1404 to 1412cm$^{-1}$ (R4) as well as 1930 to 1935 cm$^{-1}$ (R5). Around 820, 1408, and 1930 cm$^{-1}$ the imaginary part (absorption contribution) of the complex refractive index of NAT has pronounced features (Höpfner et al., 2006), whereas around 960 cm$^{-1}$ the real part (scattering contribution) of the complex refractive index of ice has a pronounced minimum (e.g. Griessbach et al., 2016). Even though in our work the ML classifiers are given BTDs (computed from radiance) as input and refractive indices are not directly used in the classification process, the latter can provide insights on microphysical properties of the different PSC particles and additional information on the features used by the ML methods.

The first and second principal components, which capture most of the variance in the data, are shown in Fig. 5. Comparing PCA and KPCA, we note that they mostly differ in terms of order and amplitude. This means that the eigenvalues change, but the eigenvectors are rather similar in the linear and non-linear case. For this dataset, the non-linear KPCA method (using a polynomial kernel) does not seem to be very sensitive to non-linear patterns that are hidden to the linear PCA method. However, it should be noted that the SVM classifier is sensitive to differences in scaling of the input features as they result from the use of PCA and KPCA for feature selection. Therefore, classification results of PCA+SVM and KPCA+SVM can still be expected to differ and are tested separately.

As discussed in Sect. 3.3, RF itself is considered to be an effective tool not only for classification but also for feature selection. It is capable of finding non-linear decision boundaries to separate between the classes. However, the method does not group the features together in components like PCA or KPCA. It is rather delivering a measure of importance of all of the

individual features. Figure 4b shows the feature importance matrix provided by the RF. Note that the values are normalized, i. e., the feature importance values of the upper triangular matrix sum up to 1. We can observe that this approach highlights similar clusters as Fig. 4a.

Similarly to PCA and KPCA, BTDs between windows in the range from 820 to 840 cm$^{-1}$ (R1) and from 956 to 964 cm$^{-1}$ (R2) are considered to be important by the RF algorithm. BTDs between 1224 to 1250 cm$^{-1}$ (R3) and 1404 to 1412 cm$^{-1}$ (R4) are also regarded as important. The importance of the RF features located in this cluster is in contrast with the relatively low BTDs variance in the same area. A similar observation can be done regarding BTDs between 782 to 800 cm$^{-1}$ and 810 to 820 cm$^{-1}$ (both belonging to R1). This region is in the range of values of the NAT feature, providing a possible explanation of the capability of the RF to detect the characteristic peak of small NAT as well as its shift with the increase of the radius. BTDs between 960 cm$^{-1}$ (R2) and 1404 to 1412 cm$^{-1}$ (R4) are also quite important. Table 3 specifically provides the most important BTDs between the different regions. Actually, Fig. 6 shows that all the windows or BTDs found here by the RF are associated with physical features of the PSC spectra, namely a peak in the real and imaginary part of the complex refractive index of NAT around 820 cm$^{-1}$ or a minimum in the real part of the complex refractive index of ice around 960 cm$^{-1}$. STS can be identified based on the absence of these features. Considering the larger windows W, the matrices of the variance and of the RF feature importance seem to agree, with the exception of W3 ($\sim$ 820 cm$^{-1}$) that is regarded as important by the RF scheme but is not characterized by high variance, confirming the capability of the RF of detecting the NAT feature.

A closer inspection reveals an interesting difference between PCA and KPCA on the one hand and RF on the other hand. Two additionally identified windows around $\sim$790 (BT index $\sim$ 10) and $\sim$1235 cm$^{-1}$ (BT index $\sim$ 90) are located at features in the imaginary part of the refractive index of ice and NAT, respectively (Höpfner et al., 2006). This latter set of BTDs is considered to have a large feature importance by the RF method but does not show a particularly large variance. This suggests that a supervised method like RF can capture important features where unsupervised methods like PCA and KPCA may fail.

## 4.2 Hyperparameter tuning and cross-validation accuracy

Concerning classification, we compared two SVM-based classifiers that take as input the features from PCA and KPCA and the RF that uses the BTD features without prior feature selection. The first step in applying the classifiers is training and tuning of the hyperparameters. Cross-validation is a standard method to find optimal hyperparameters and to validate a ML model (Kohavi, 1995). For cross-validation the dataset is split in a number of subsets, called folds. The model is trained on all the folds, except for one, which is used for testing. This procedure is repeated until the model has been tested on all the folds. The cross-validation accuracy refers to the mean error of the classification results for the testing data sets. Cross-validation is considered essential to avoid overfitting while training a ML model. Selecting the best hyperparameters that maximize the cross-validation accuracy of a ML model is of great importance to exploit the models capabilities at a maximum.

In this study, we applied 5-fold cross-validation on the CSDB dataset. For the SVM models we decided to utilize a grid-search approach to find the hyperparameters. As the parameter space of the RF model is much larger, a random-search approach was adopted (Bergstra and Bengio, 2012). The test values and optimum values of the hyperparameters for the SVM and RF classifiers are reported in Tables 4 and 5, respectively. For the optimum hyperparameter values, all classification methods

provided an overall prediction accuracy close to 99%. Also, our tests showed that the ML methods considered here for the PSC classification problem are rather robust against changes of the hyperparameters.

During the training of the classifiers, we conducted two experiments. In the first experiment, we checked how large the amount of synthetic samples from the CSDB needs to be in order to obtain good cross-validation accuracy. For this experiment, we performed the training with subsets of the original CSDB data, using randomly sampled fractions of 50%, 20%, 10%, 5%, 2%, 1%, 0.05%, 0.02%, 0.01%, 0.005%, 0.002%, and 0.001% of the full dataset. This experiment has been run for all three ML models (PCA+SVM, KPCA+SVM, and RF) using the optimal hyperparameters found during the cross-validation step. The results in Fig. 7 show that using even substantially smaller datasets ($> 0.02\%$ of the original data or about 1200 samples) would still result in acceptable prediction accuracy ($> 80\%$). This result is surprising and points to a potential limitation of the CSDB for the purpose of training ML models that will be discussed in more detail in Sect. 5.

In the second experiment, we intentionally performed and analyzed the training and testing of the RF method with a rather small subset of data. Although the results from this procedure are less robust, they can help pinpoint potential issues that cannot be detected using the full data set. We computed different scores to assess the quality of the prediction for the RF classifier in the case of 600 randomly selected samples used for training and around 200 samples used for testing. As shown in Table 6, also using a limited number of samples for training leads to very high classification accuracy. The metrics used in Table 6 are precision $P = TP/(TP + FP)$, recall $R = TP/(TP + FN)$, and f1-score $F1 = 2(R \times P)/(R + P)$, where $TP$ is the number of true positives, $FP$ the number of false positives, $FN$ the number of false negatives, and support is the number of samples (Tharwat, 2018). As reported in Table 6, it is found that ice and small NAT accuracies are higher than the ones of STS. This is a hint to the fact that distinguishing small NAT and ice from the other classes is an easier task than separating spectra of PSC containing larger NAT particles from those populated with STS, which is consistent with previous studies (Höpfner et al., 2009).

An additional experiment was performed on the CSDB spectra labeled as large NAT. The BC misclassifies a large amount of those spectra (99% of them classified as STS_mix), whereas the proposed ML methods correctly classify them as large NAT (Tab. 7). This experiment suggests that the new classification schemes can help in overcoming the inability of the BC in discriminating between large NAT and STS.

## 4.3 Classification using real MIPAS data

### 4.3.1 Case studies

For three case studies looking at individual days of MIPAS observations, two in the Southern Hemisphere and one in the Northern Hemisphere winter season, we compared the results of the different classification methods (Figs. 8 to 10). Early in the Southern Hemisphere PSC season, on 14 June 2009 (Fig. 8), we found that the classification results are mostly coherent among all the classifiers, not only from a quantitative point of view but also geographically, especially concerning the separation of ice and STS PSCs. Further, we found that most of the PSCs, which were labeled as NAT by the Bayesian classifier, were classified as STS by the ML classification methods. While both SVM classification schemes did not indicate the presence of

NAT, the RF found some NAT, but mostly at different places than the Bayesian classifier. Note that from a climatological point of view, NAT PSCs are not expected to be the dominant PSC type until mid to end of June for the Southern Hemisphere (Pitts et al., 2018).

Later in the Southern Hemisphere PSC season, on 26 August 2009 (Fig. 9), it is again found that the separation between ice and non-ice PSCs is largely consistent for all the classifiers. The NAT predictions by the RF classifier tend to agree better with the Bayesian classifier than the NAT classifications by the SVM method. Overall, the Southern Hemisphere case studies seem to suggest that the SVM classifiers (using PCA or KPCA) underestimate the presence of NAT PSCs compared to the BC and the RF classifiers. We note that separating the NAT and STS classes from limb infrared spectra presents some difficulties.

As a third case study, we analyzed classification results for 25 January 2007 for the Northern Hemisphere (Fig. 10). This case was already analyzed to some extent by Hoffmann et al. (2017). It is considered to be particularly interesting, as ice PSCs have been detected over Scandinavia at synoptic-scale temperatures well above the frost point. Hoffmann et al. (2017) provided evidence that the PSC formation in this case was triggered by orographic gravity waves over the Scandinavian Mountains. Also in this case study the classification of ice PSCs over Scandinvia shows a good agreement for the new ML methods with the Bayesian classifier. Further, we see that the two SVM and the RF methods identified small NAT where the Bayesian classifier also found NAT. However, at the locations, where the Bayesian classifier indicates a mixture of NAT and STS the ML methods indicate STS, and the ML methods indicate large NAT at locations where the Bayesian classifier found STS.

### 4.3.2 Seasonal analyses

For a seasonal analysis, we first considered MIPAS observations during the months from May to September 2009. Figures 11 to 13 show the area coverage for each class of PSC along time and altitude. Comparing the time series of the classification results, we can assess the agreement quantitatively. The mixed composition classes of the Bayesian classifier (NAT_STS, ICE_STS and ICE_NAT) are not considered in this analysis. Taking a look at STS (Fig. 11), all the classifiers predict an early season appearance. While the RF predicts a time series that resembles quite closely the one predicted by the Bayesian classifier, the other two ML methods (PCA+SVM and KPCA+SVM) predict a significantly larger coverage of STS clouds over the winter. Regarding the ice PSCs (Fig. 12), the patterns in the time series are similar between all classifiers. However, we can observe that even if the spatiotemporal characteristics are similar, both SVM methods predict a notably larger area covered by ice clouds. Moreover, the KPCA+SVM classifier predicts an earlier emergence of ice with respect to the other classifiers. Considering the NAT time series (Fig. 13), all the classifiers predict a late appearance during the season. The classification schemes based on SVM predict a much lower presence of NAT with respect to the RF and the Bayesian classifier. Furthermore, most of the bins with a high value of NAT coverage in the Bayesian classification scheme are predicted as small NAT particles. This result confirms that the spectral features of small NAT are strong enough to find a good decision boundary, as explained in Sect. 2.2.

Figure 14 shows the overall percentages of the PSC classes for May to September 2009 for the Southern Hemisphere. The occurrence frequencies of ice PSCs are quite consistent ranging from 32 % for the Bayesian classifier to 39 % for KPCA+SVM. It is found that the approaches based on SVM slightly overestimate the presence of ice with respect to the RF (35 %) and the Bayesian classifier. However, the main differences that were encountered are in the separation between STS and NAT. The

two classification schemes using SVM predict a much smaller amount of NAT PSCs (17 and 26 % taking small and large NAT together) compared to the RF (33 % considering only small NAT, 37 % taking small and large NAT together) and the Bayesian classifier (32 % NAT). The RF and the Bayesian classifier are more coherent between themselves. Other interesting findings are related to the classification between small and large NAT. Indeed, the vast majority of the NAT predictions in the KPCA+SVM and RF methods belong to the small NAT class. PCA+SVM diverges significantly from the other methods, largely underestimating small NAT and overestimating large NAT. This suggests once more that the discrimination between small NAT and STS PSCs is more easily possible using mid-infrared spectra for classification, while larger NAT PSCs are harder to separate.

In addition to the results presented above, we conducted the seasonal analyses also for MIPAS observations acquired in the months from November 2006 to February 2007 in the Northern Hemisphere (Fig. 15). As expected, a much smaller fraction of ice PSCs $(4-6\,\%)$ has been found compared to the Southern Hemisphere. As in the Southern Hemisphere winter, the SVM classifiers taking as input the PCA and KPCA features found significantly less NAT (both 6 %) than the Bayesian classifier (15 %), whereas the RF classifier identified a significantly larger fraction of large NAT spectra (30 %) that resulted in a significantly higher NAT detection rate (37 %). This finding may point to a potential improvement of the RF classifier compared to the Bayesian classifier. In fact, it had been already reported by Spang et al. (2016) that the Bayesian classifier for MIPAS underestimated the fraction of NAT clouds compared to Cloud-Aerosol LIDAR with Orthogonal Polarization (CALIOP) observations. Further, the STS partitioning between the three STS subclasses is different between the Southern and Northern Hemisphere winters. While in the Southern Hemisphere STS 1 is dominating, in the Northern Hemisphere STS 2 is dominating and the fraction of STS 3 is significantly increased. This result is plausible, because the Northern Hemisphere winters are warmer than the Southern Hemisphere winters, and STS 1 forms at lower temperatures (e.g. $\sim189\,\text{K}$) than STS 2 ($\sim192\,\text{K}$) and STS 3 ($\sim195\,\text{K}$ at 50 hPa, Carslaw et al. (1995)).

Figures 16 and 2 show cross tabulations between the classification results of the Bayesian classifier and the three ML methods. They allow us to directly assess how much the different classification schemes agree in terms of their predictions for the different classes. For instance, considering the ice class of the PCA+SVM and KPCA+SVM classifiers, it can be seen that around 80% of the samples were classified consistently with the Bayesian method, while this percentage is above 90% for the RF (Fig. 16). Concerning NAT, the RF classifier predicts as small NAT more than 80% of what had been classified as NAT class by the Bayesian classifier (Fig. 2). The PCA+SVM and KPCA+SVM methods predict a smaller fraction of small NAT for the NAT class of the Bayesian classifier, around 30% and 70%, respectively. The PCA+SVM in particular is predicting a significantly smaller amount of samples belonging to the small NAT class than the other methods (Fig. 16), while it predicts a larger number of samples of the STS subclasses. This result may suggest that PCA+SVM and KPCA+SVM are not as sensitive as BC for small NAT detection, while RF is. Considering the STS subclasses of the RF and KPCA+SVM classifiers altogether, they seem to mostly agree with the STS_mix predictions of the Bayesian classifier. On the other hand, the total number of samples predicted by the PCA+SVM scheme as belonging to the STS subclasses is notably larger than the predictions of the Bayesian classifier (Fig. 16). This finding is in line with what has been discussed a few lines above and in Sect. 4.3.2. There is a large percentage of spectra predicted as large NAT by the proposed ML methods that are instead classified as STS by the BC,

especially in the results of the RF scheme. This is probably caused by the fact that the BC misclassifies spectra of large NAT, as discussed in Sect. 4.2 for the CSDB.

## 5   Summary and conclusions

In this study, we investigated whether ML methods can be applied for the PSC classification of infrared limb spectra. We compared the classification results obtained by three different ML methods – PCA+SVM, KPCA+SVM, and RF – with those of the Bayesian classifier introduced by Spang et al. (2016). First, we discussed PCA, KPCA, and RF as methods for feature extraction from mid-infrared spectral regions and showed that the selected features correspond with distinct features in the complex refractive indices of NAT and ice PSCs. Then we compared classification results obtained by the ML methods with respect to previous work using conventional classification methods combined with a Bayesian approach.

We presented three case studies as well as seasonal analyses for the validation and comparison of the classification results. Based on the case studies, we showed that there is spatial agreement of the ML method predictions between ice and non-ice PSCs. However, there is some disagreement between NAT and STS. We evaluated time series and pie charts of cloud coverage for the Southern Hemisphere polar winter 2009 and the Northern Hemisphere polar winter 2006/2007, showing that all methods are highly consistent with respect to the classification of ice. For the NAT and STS predictions, RF and the Bayesian classifier tend to agree best, whereas the SVM methods yielded larger differences. The agreement between the different classification schemes was further quantified by means of cross tabulation. While the SVM methods found significantly less NAT than the Bayesian classifier, the RF classifier found slightly more NAT than the Bayesian classifier. The RF results might be more realistic, because the Bayesian classifier is known to find less NAT for MIPAS compared to CALIOP satellite observations, especially for Northern Hemisphere winter conditions (Spang et al., 2016). A practical advantage of RF presented in Sect. 3.3 and further discussed in 4.1, is that it enables a better control on the importance of the features it selects to train the model. Moreover, RF is a fully supervised method, from feature selection to training, whereas the feature extraction methods PCA and KPCA are unsupervised methods and may fail to capture important features if they do not show high variance. From the user point of view, RF is also simpler to deploy since it embeds feature selection and does not require a two-step process of feature extraction and training (unlike PCA+SVM and KPCA+SVM). Parallel implementations of the ML methods presented in this paper are also available, enabling significant acceleration of model training and prediction with large amount of data (Cavallaro et al., 2015; Genuer et al., 2017).

The Bayesian method developed by Spang et al. (2016) requires a priori knowledge of a domain expert to select the decision boundaries and to tune the probabilities used for classification for different areas in the feature space. The ML schemes proposed in this work are more objective in the premises and rely only on the available training data without additional assumptions. Models have been trained on the CSDB, a simulation dataset that has been created systematically sampling the parameter space, not reflecting the natural occurrence frequencies of parameters. This point is in our opinion of great importance, as we demonstrated that ML methods are capable of predicting PSC composition classes without the need of substantial prior knowledge, providing a mean for consistency checking of subjective assessments. Although the lack of ground truth nar-

rows the assessment down to comparison with other classification schemes, we found that the classification results of the ML methods are consistent with spectral features of the PSC particles, in particular, the features found in the real and imaginary part of their refractive indices. Another important benefit of the proposed ML methods is that they have shown the potential of extending the prediction to NAT particles with large radius, which was not possible with the BC scheme. This aspect has been

successfully tested on the synthetic CSDB dataset and might be a promising path for future research.

However, there are still some limitations to the proposed ML approach. First, the feature selection methods found the highest variance and feature importance at spectral windows where ice and NAT have pronounced features in the complex refractive indices, whereas the main features of STS are located at wavenumbers not covered by the CSDB. Since the classification of STS is therefore based on the absence of features in the optical properties and for the large NAT particles the features in the

optical properties vanish as well, the discrimination between STS and large NAT is more complicated than the identification of ice. Hence, we suppose that the inclusion of more spectral windows, especially regions where the optical properties of STS have features, may bear the potential to improve the separation between STS and NAT. Second, we showed that using a much smaller subset of the original CSDB for training of the ML methods would have been sufficient to achieve similar classification results. This suggests that the information provided by the CSDB is largely redundant, at least in terms of training of the ML

methods. Despite the fact that the CSDB contains many training spectra, it was calculated only for a limited number of PSC volume densities, particle sizes, and cloud layer heights and depths as well as fixed atmospheric background conditions. It could be helpful to test the ML methods using a training data set providing better coverage of the micro- and macrophysical parameter space and more variability in the atmospheric background conditions. Third, in the CSDB and the ML classification schemes we assumed only pure constituent (ice, NAT, STS 1, STS 2, STS 3) PSCs, whereas in the atmosphere mixed clouds

are frequently observed (e.g. Deshler et al., 2003; Pitts et al., 2018). In future work, mixed PSCs should be included, as an investigation of mixed PSCs could be beneficial to assess how far the ML methods applied to limb infrared spectra agree with predictions from CALIOP measurements that already comprise mixed type scenarios.

In general, the presented classification methods are straightforward to adopt on spectrally resolved measurements of other infrared limb sensors like the CRyogenic Infrared Spectrometers and Telescopes for the Atmosphere (CRISTA) (Offermann

et al., 1999) or the GLObal limb Radiance Imager for the Atmosphere (GLORIA) (Riese et al., 2005; Ungermann et al., 2010; Riese et al., 2014) space- or airborne instruments. It could be of interest to extend the methods to combine different observational datasets, even with different types of sensors providing different spectral and geometrical properties of their acquisitions. This study has assessed the potential of ML methods in predicting PSC composition classes, which may be a starting point for new classification schemes for different aerosol types in the upper troposphere and lower stratosphere region

(Sembhi et al., 2012; Griessbach et al., 2014, 2016), helping to answer open questions about the role of these particles in the atmospheric radiation budget.

*Code and data availability.* The MIPAS Level-1B IPF version 7.11 data can be accessed via ESA's Earth Online portal at https://earth.esa.int/ (last access: 10 December 2019). The CSDB database can be obtained by contacting Michael Höpfner, Karlsruhe. The software repository containing the ML codes developed for this study is available at https://gitlab.com/rocco.sedona/psc_mipas_classification.

*Author contributions.* GC, LH, and RSp developed the concept for this study. RoS developed the software and conducted the formal analysis
5  of the results. SG, MH, and RSp provided expertise on the MIPAS measurements. MH prepared and provided the CSDB. GC and MR provided expertise on the ML methods. RoS wrote the manuscript with contributions from all co-authors.

*Competing interests.* The authors declare that they have no conflict of interest.

*Acknowledgements.* We thank the European Space Agency (ESA) for making the Envisat MIPAS data available. We found the scikit-learn software package (https://scikit-learn.org/, last access: 10 December 2019) of great importance for the development of the code for this study.

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

**Table 1.** Infrared spectral regions considered for PSC classification.

| Spectral region | Index range | Wavenumber range [cm$^{-1}$] |
|:---:|:---:|:---:|
| R1 | 0 – 57 | 782 – 840 |
| R2 | 58 – 83 | 940 – 965 |
| R3 | 84 – 98 | 1224 – 1250 |
| R4 | 99 – 106 | 1404 – 1412 |
| R5 | 107 – 112 | 1930 – 1935 |
| R6 | 113 – 125 | 1972 – 1985 |
| R7 | 126 – 130 | 2001 – 2006 |
| R8 | 131 – 136 | 2140 – 2146 |
| W1 | 137 | 788.2 – 796.2 |
| W2 | 138 | 832 – 834.4 |
| W3 | 139 | 819 – 821 |
| W4 | 140 | 832.3 – 834.4 |
| W5 | 141 | 947.5 – 950 |

**Table 2.** PSC constituents, particle concentrations, and sizes covered by the CSDB.

| PSC constituents | Volume density [$\mu m^3 cm^{-3}$] | Median radius [$\mu m$] |
|:---:|:---:|:---:|
| ice | 10, 50, 100 | 1.0, 2.0, 3.0, 4.0, 5.0, 10.0 |
| NAT | 0.1, 0.5, 1.0, 5.0, 10.0 | 0.5, 1.0, 2.0, 3.0, 4.0, 5.0 |
| STS | 0.1, 0.5, 1.0, 5.0, 10.0 | 0.1, 0.5, 1.0 |

**Table 3.** Top ten list of BTDs providing maximum feature importance as estimated by the RF classifier.

| Feature importance | BTD indices | BTD wavenumbers [cm$^{-1}$] |
|:---:|:---:|:---:|
| 0.006815 | 85 – 105 | 1225.5 – 1410.5 |
| 0.005798 | 61 – 83 | 942.5 – 964.5 |
| 0.004334 | 57 – 76 | 839.5 – 957.5 |
| 0.003233 | 37 – 56 | 819.5 – 838.5 |
| 0.002649 | 86 – 139 | 1226.5 – 820 |
| 0.002633 | 58 – 139 | 840.5 – 820 |
| 0.002272 | 40 – 87 | 822.5 – 1227.5 |
| 0.001677 | 26 – 139 | 808.5 – 820 |
| 0.001592 | 27 – 101 | 809.5 – 1406.5 |
| 0.001033 | 102 – 137 | 1407.5 – 792.2 |

**Table 4.** Hyperparameter choices considered for the SVM classifier.

| Hyperparameter | Test values | Optimal value |
|:---:|:---:|:---:|
| kernel | linear, RBF, polynomial | RBF |
| $C$ | 1, 10, 100, 1000 | 1000 |
| $\gamma$ | 0.0001, 0.001, 0.01, 0.1, 1, 10 | 1 (PCA) / 10 (KPCA) |

**Table 5.** Hyperparameter choices considered for the RF classifier.

| Hyperparameter | Test values | Optimal value |
|---|---|---|
| number of estimators | $200, 210, \ldots, 2000$ | 1000 |
| maximum number of features | auto, sqrt | auto |
| maximum depth | $10, 20, \ldots, 110$ | 50 |
| minimum number of samples before split | 2, 5, 10 | 2 |
| minimum number of samples per leaf | 1, 2, 4 | 1 |
| bootstrapping | true, false | false |

**Table 6.** Scores of the RF classifier on a small subset of CSDB samples.

| Class | Precision | Recall | f1-score | Support |
|---|---|---|---|---|
| ice | 1.00 | 1.00 | 1.00 | 56 |
| NAT_large | 1.00 | 0.91 | 0.95 | 23 |
| NAT_small | 1.00 | 1.00 | 1.00 | 33 |
| STS_1 | 0.96 | 0.76 | 0.85 | 34 |
| STS_2 | 0.78 | 0.97 | 0.86 | 33 |
| STS_3 | 0.94 | 0.97 | 0.96 | 34 |
| total | 0.95 | 0.94 | 0.94 | 210 |

**Table 7.** Predicted labels vs CSDB classes, analysis restricted to NAT large (radius >2 μm).

| NAT large, CSDB | | | | | |
|---|---|---|---|---|---|
| BC Class | pred. by BC | proposed ML Class | pred. by PCA+SVM | pred. by KPCA+SVM | pred. by RF |
| ICE | 0 | ICE | 0 | 0 | 0 |
| NAT | 0.0012 | NAT_small | 0 | 0 | 0 |
| | | NAT_large | 1 | 1 | 1 |
| STS_mix | 0.9988 | STS_1 | 0 | 0 | 0 |
| | | STS_2 | 0 | 0 | 0 |
| | | STS_3 | 0 | 0 | 0 |
| NAT \STS | 0 | | | | |
| ICE \NAT | 0 | | | | |
| ICE \STS | 0 | | | | |

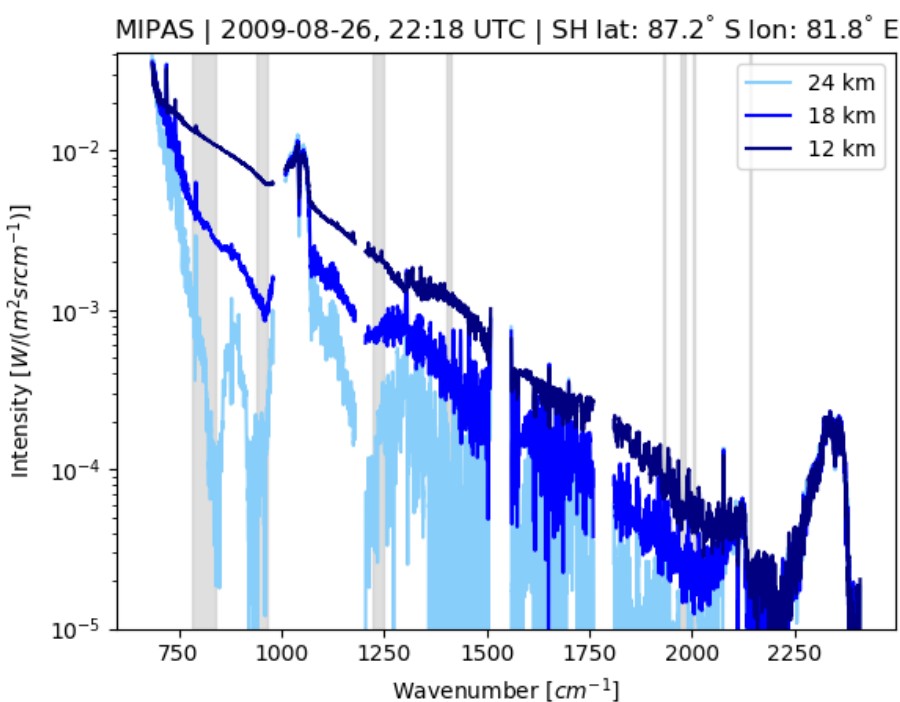

**Figure 1.** MIPAS measurements in Southern Hemisphere polar winter at three tangent altitudes from the same profile showing clear air (light blue), optically thin (blue), and optically thick (dark blue) conditions. The grey bars indicate the wavenumber regions considered for PSC classification in this study.

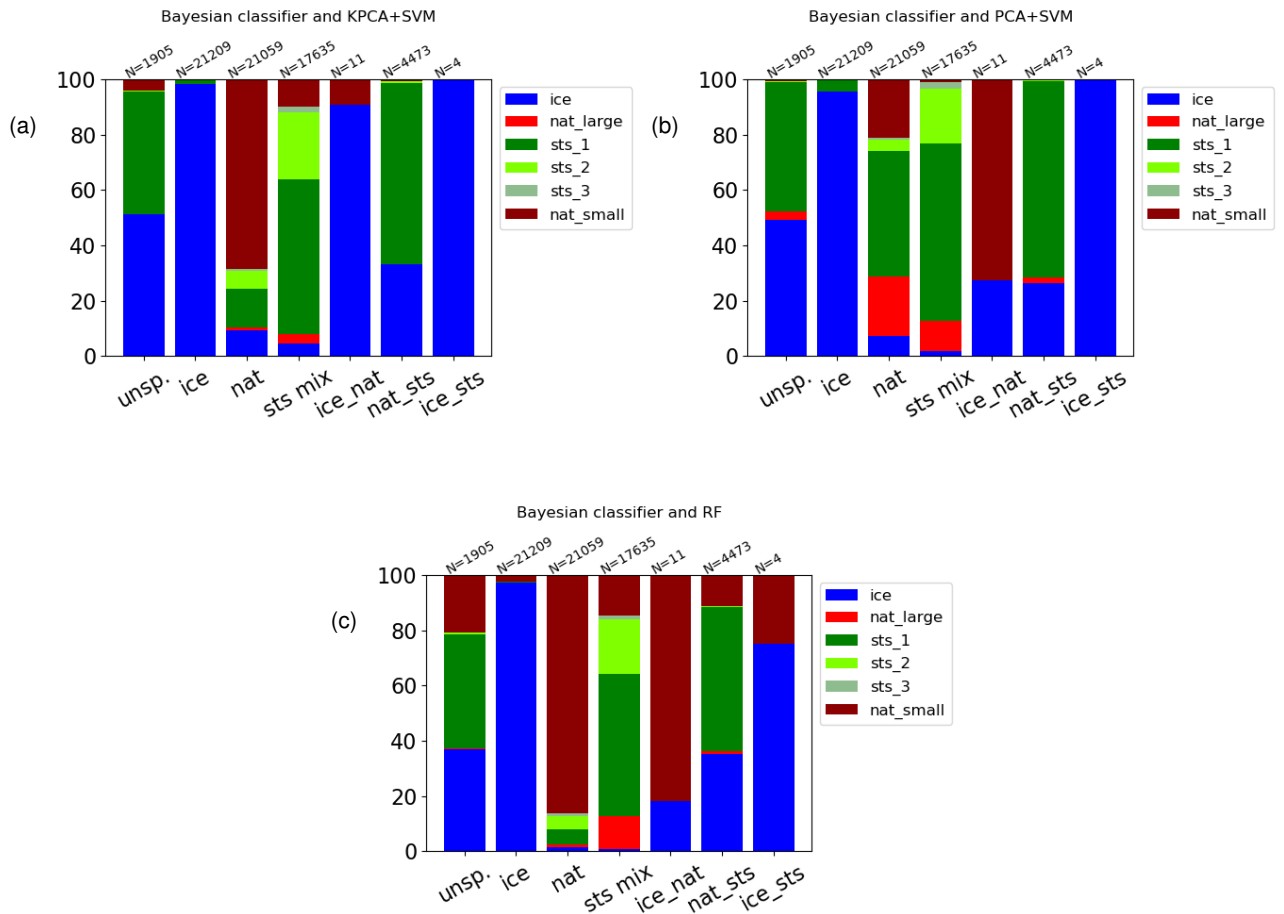

**Figure 2.** Intercomparison of ML and Bayesian classifiers for Southern Hemisphere winter (May to September 2009). Ticks on the x-axis ticks represent the classes of the BC. The y-axis indicates the fraction of the classes as predicted by the KPCA+SVM (a), PCA+SVM classifier (b), and the RF classifier (c). N is the number of samples belonging to each class of the Bayesian classifier.

(a)

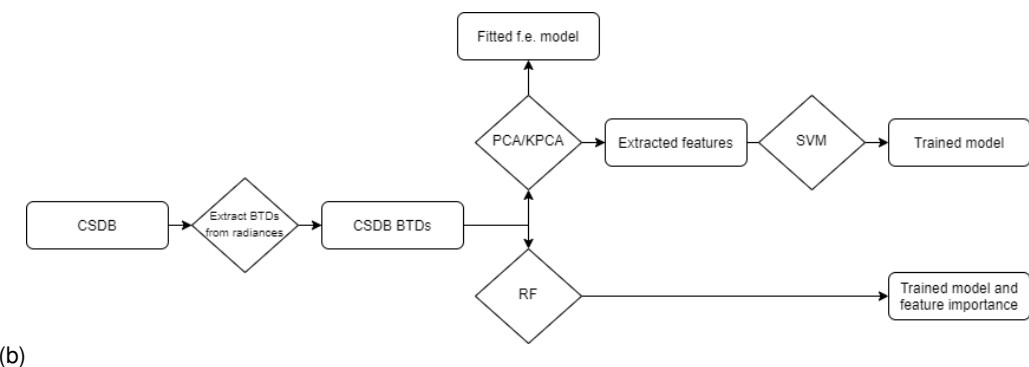

(b)

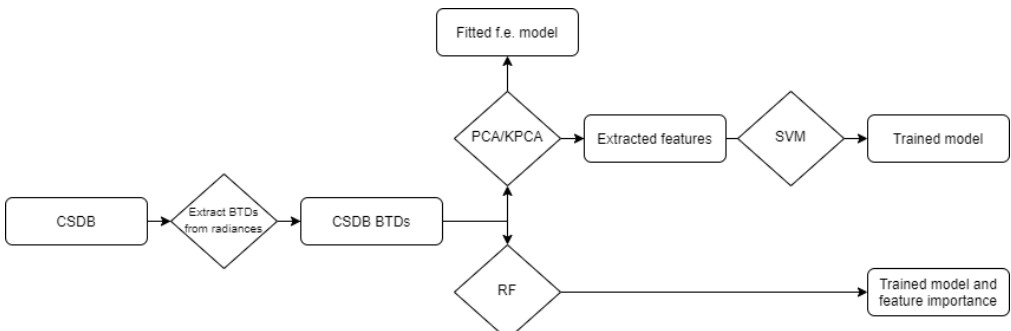

**Figure 3.** (a) Flowchart of the training process and (b) prediction. "F.e." stays for feature extraction.

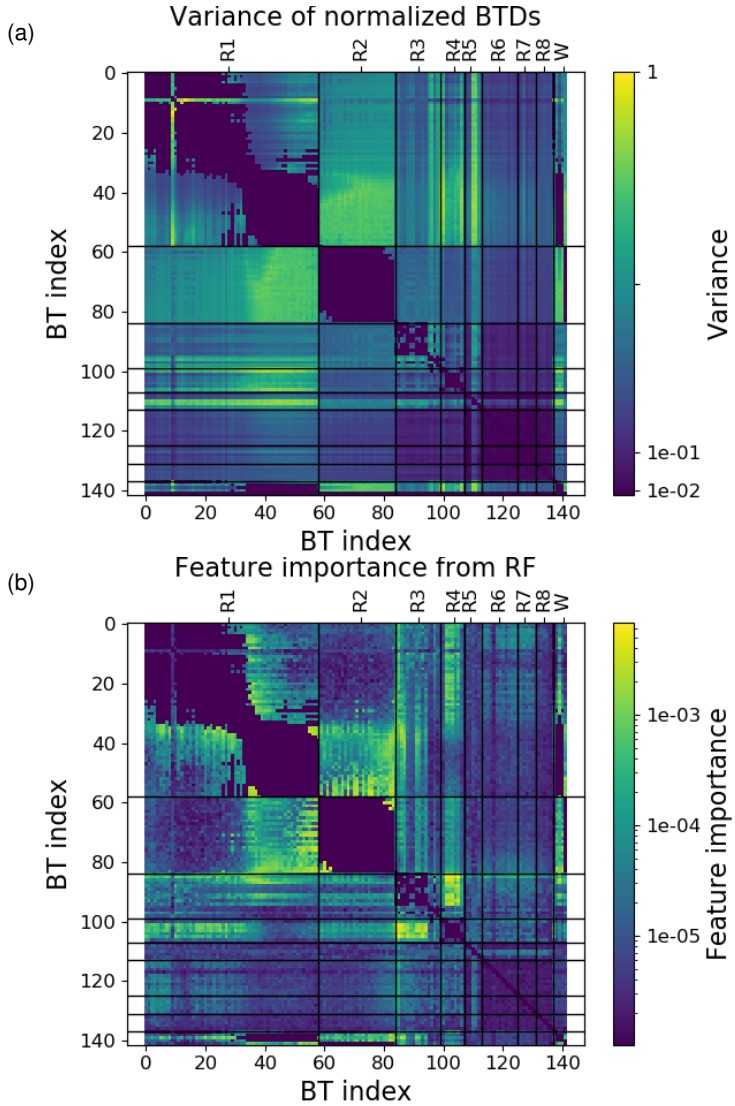

**Figure 4.** Variance of normalized BTDs (a) and feature importance as estimated by the RF classifier (b). The BT index numbers on the x- and y-axis correspond to the spectral regions as listed in Table 1.

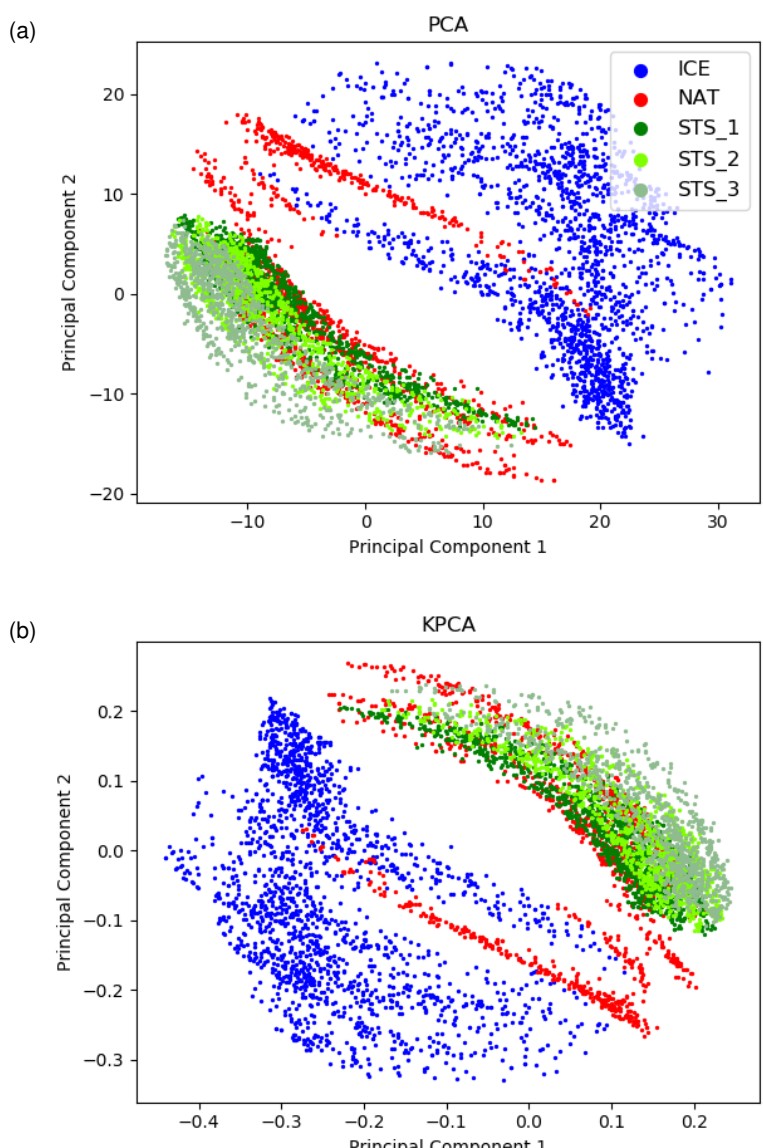

**Figure 5.** Correlations of the first two principal components from the PCA (a) and KPCA (b) analysis applied to the CSDB.

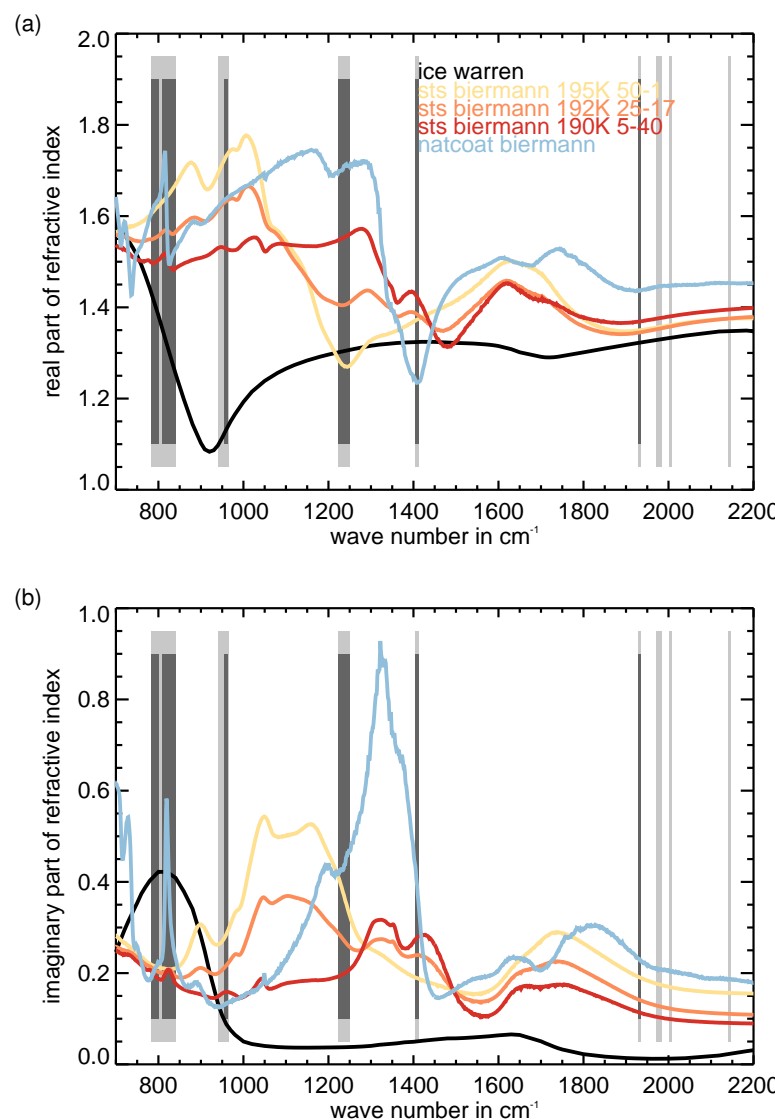

**Figure 6.** Real (a) and imaginary (b) part of PSC particle refractive indices. The grey bars represent the 8 spectral regions considered in this study.

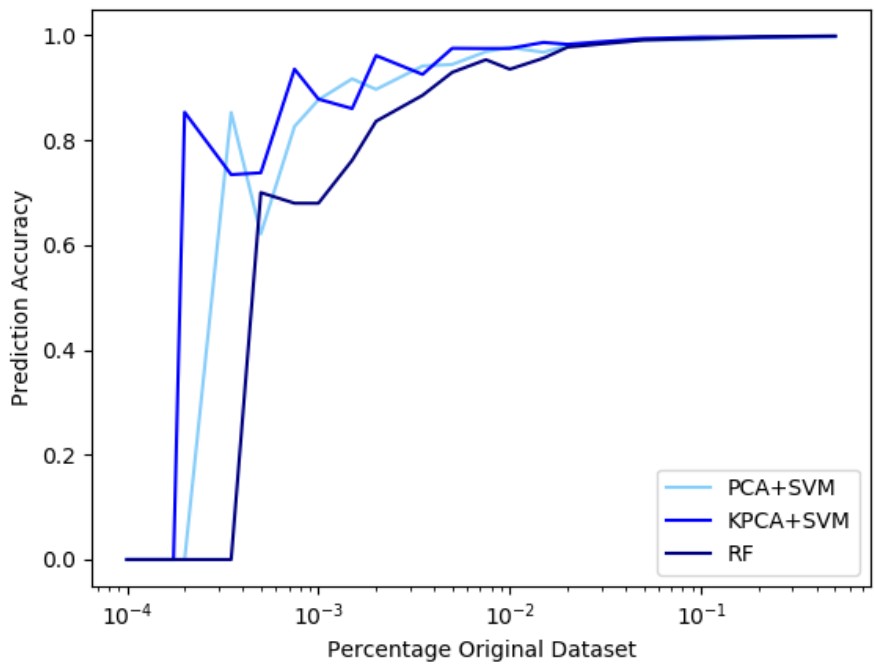

**Figure 7.** Prediction accuracy using subsets of the CSDB of different size.

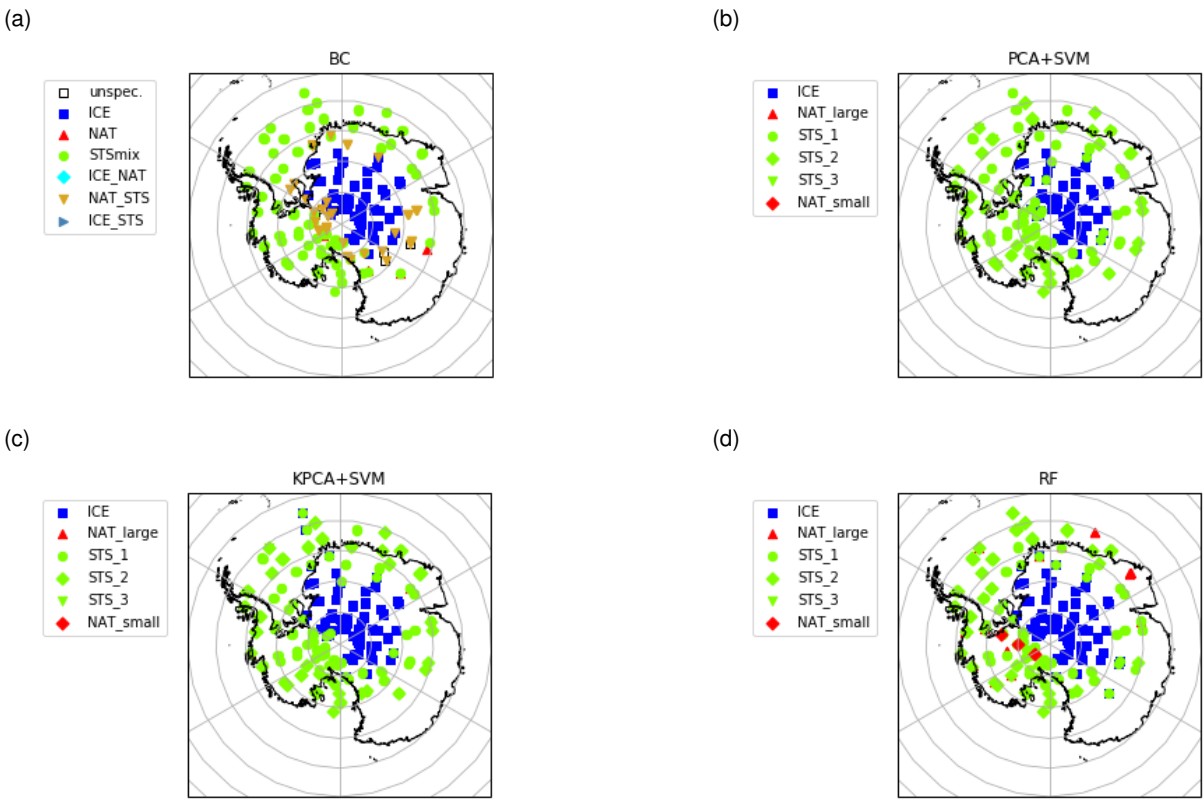

**Figure 8.** MIPAS observations of PSCs on 14 June 2009 in the Southern Hemisphere at tangent altitudes between 18 and 22 km. The classification was performed with (a) the Bayesian classifier, (b) the SVM based on PCA features, (c) the SVM based on KPCA features, and (d) the RF classifier.

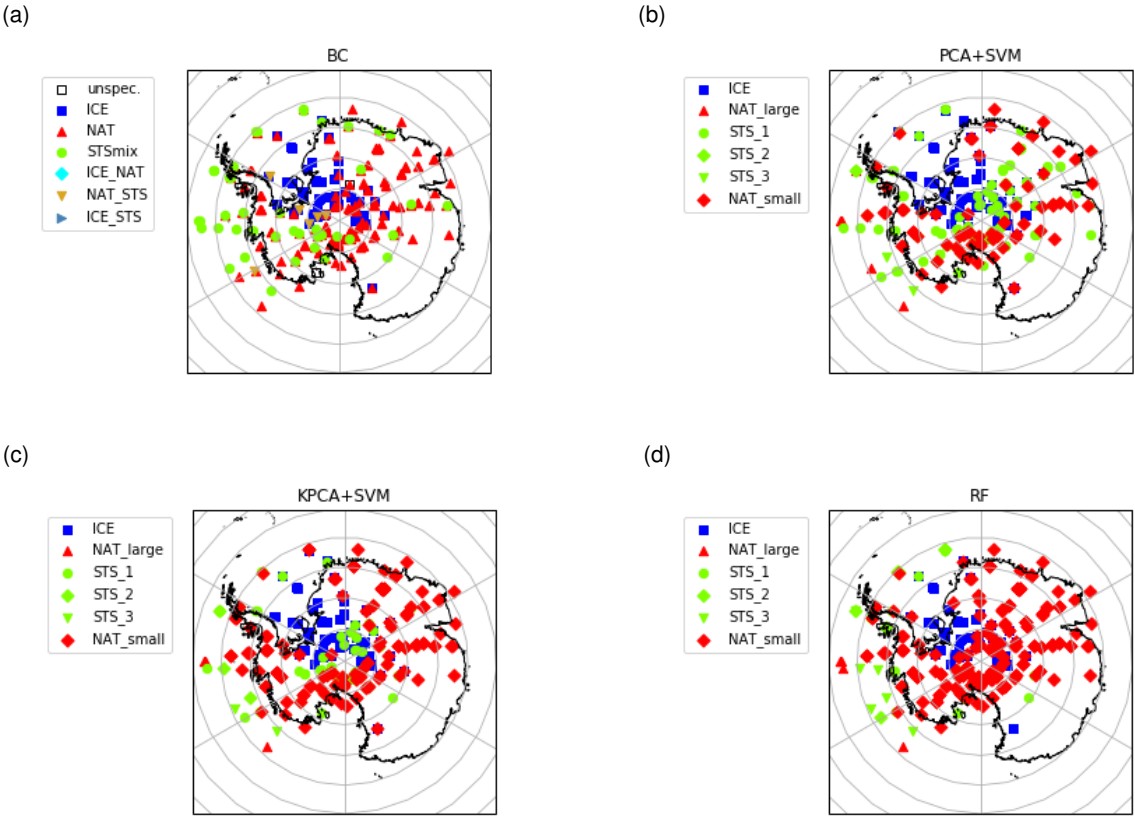

**Figure 9.** Same as Fig. 8, but for 26 August 2009.

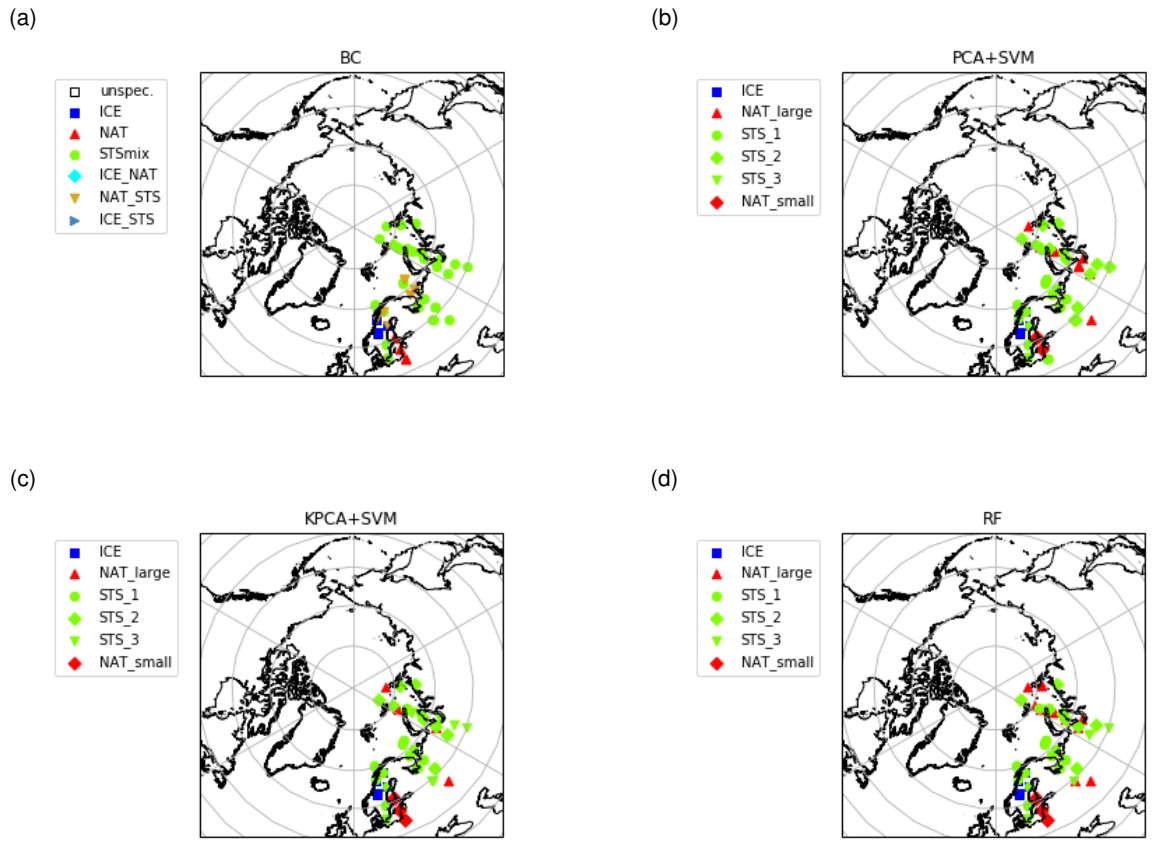

**Figure 10.** Same as Fig. 8, but for 25 January 2007 and the Northern Hemisphere.

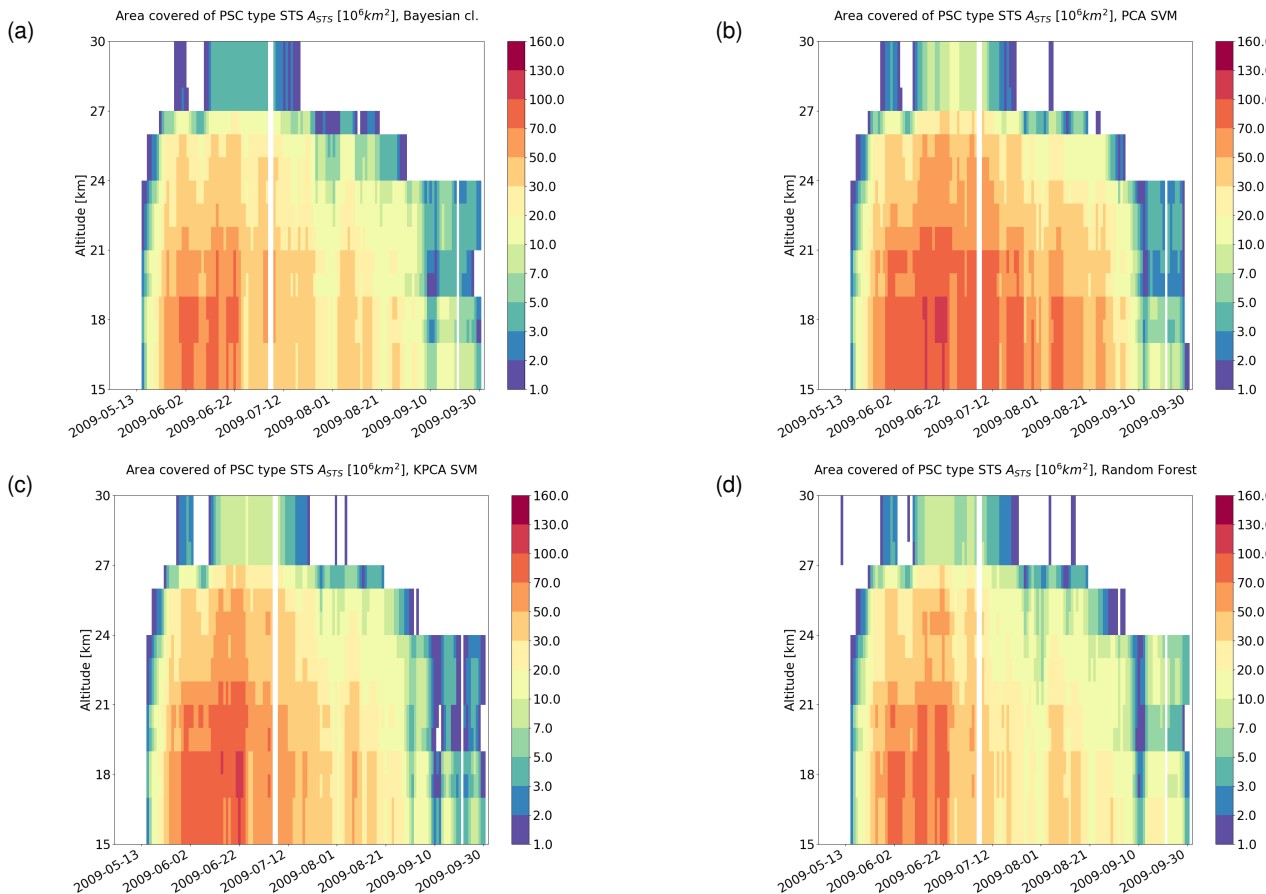

**Figure 11.** Area covered by STS clouds from May to September 2009 in the Southern Hemisphere based on results of (a) the Bayesian classifier, (b) the PCA+SVM classifier, (c) the KPCA+SVM classifier, and (d) the RF classifier. The bins span a length of 1 day in time and 1 km in altitude. A horizontal (3 days) and vertical (3 km) moving average has been applied for the sake of a smoother representation.

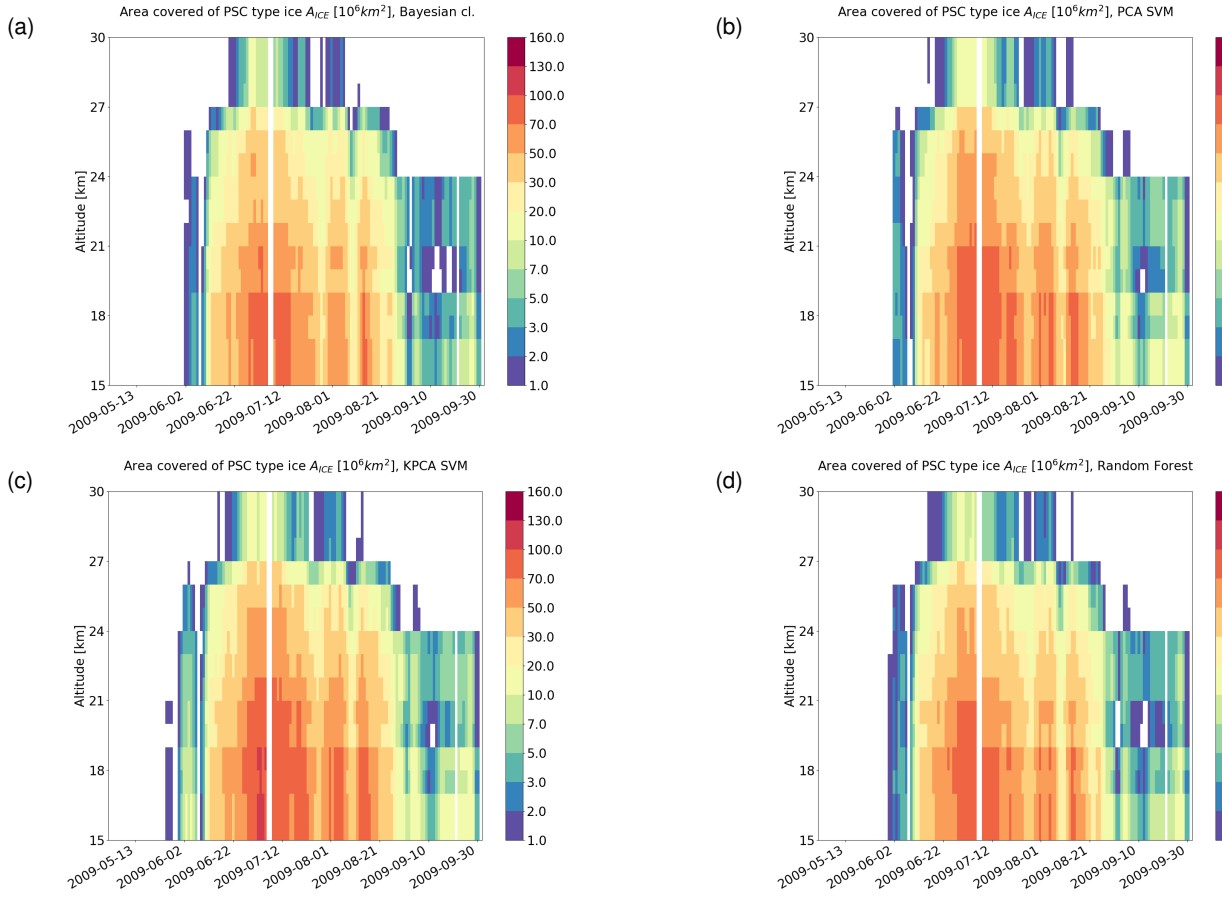

**Figure 12.** Same as Fig. 11, but for ice.

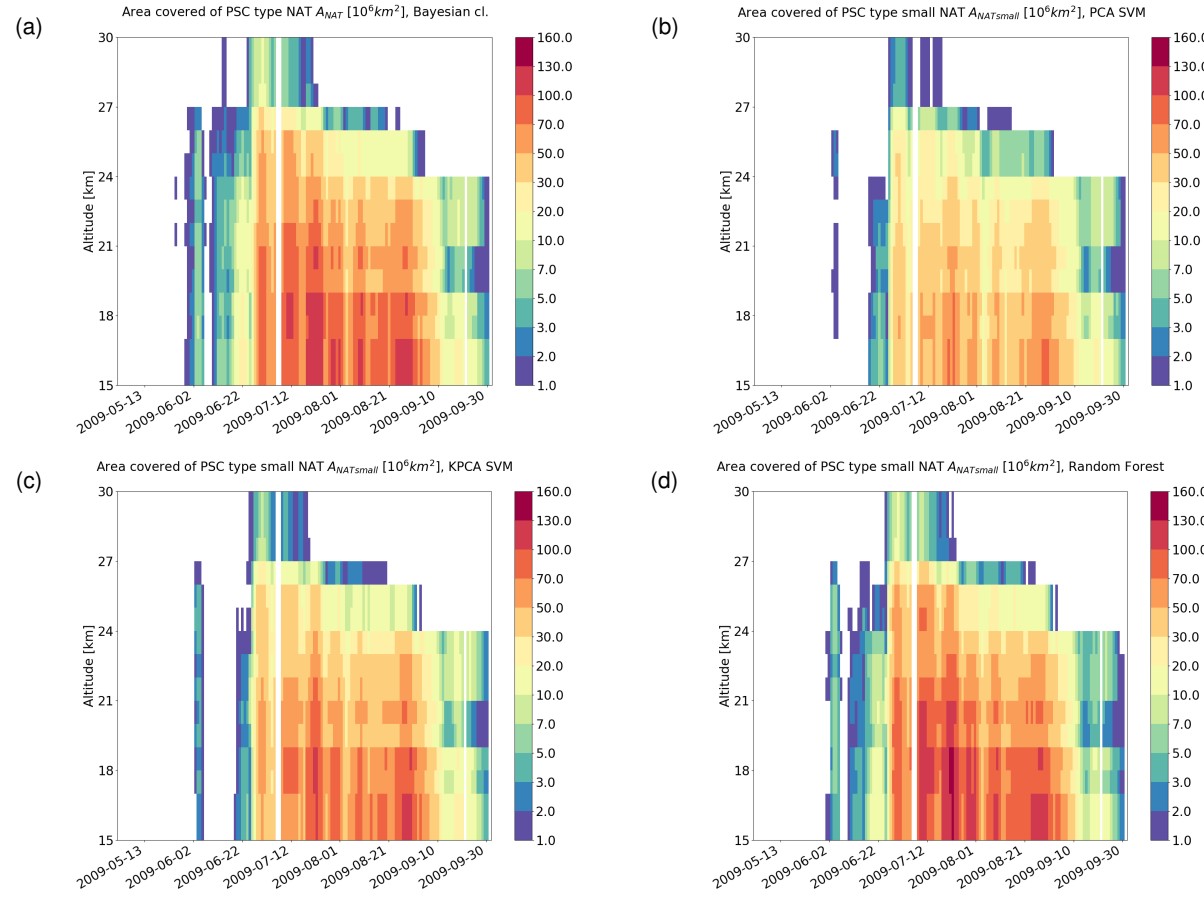

**Figure 13.** Same as Fig. 11, but for NAT.

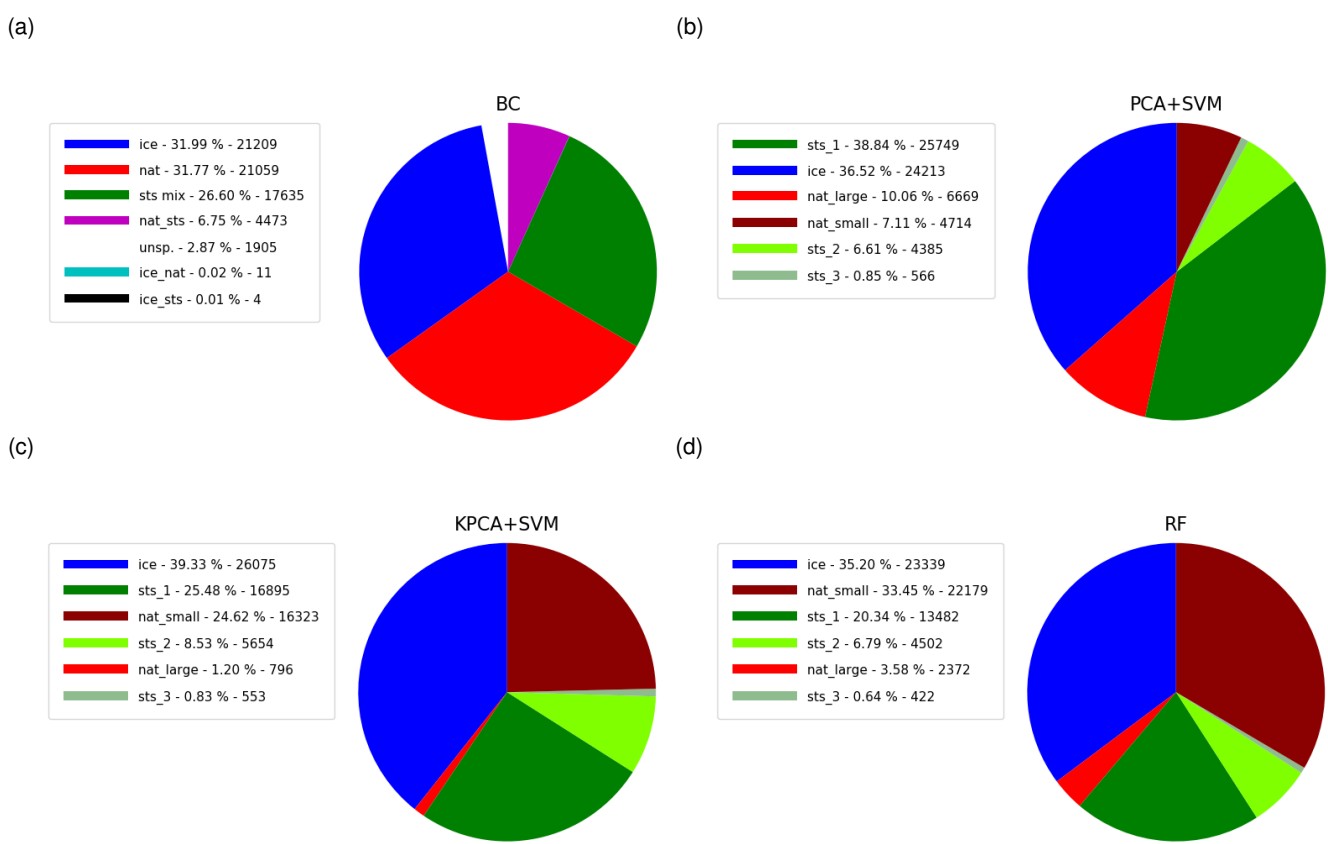

**Figure 14.** Partitioning of the PSC composition classes for the Southern Hemisphere winter (May to September 2009) derived by (a) the Bayesian classifier, (b) the PCA+SVM classifier, (c) the KPCA+SVM classifier, and (d) the RF classifier. Percentage values and number of events are reported in the legends.

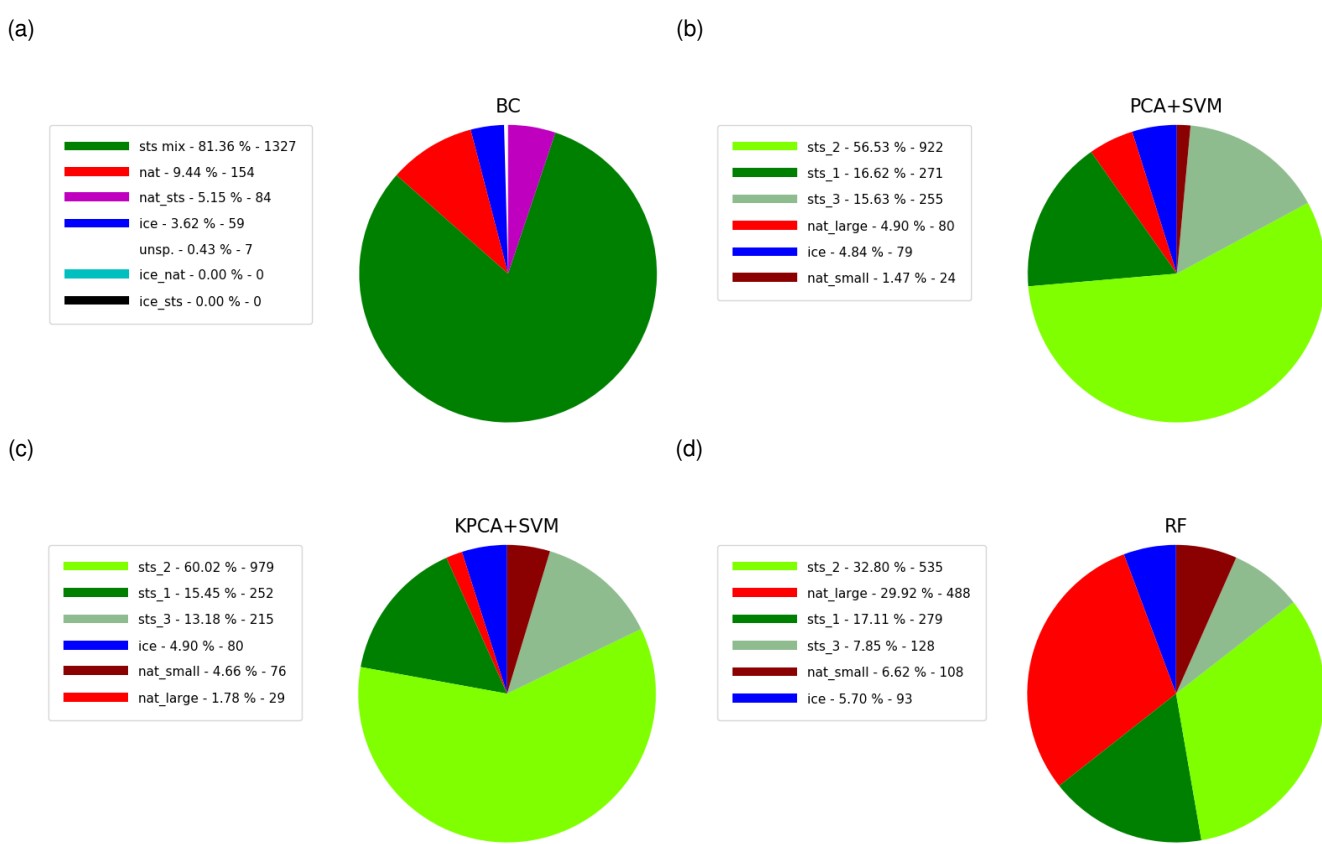

**Figure 15.** Same as Fig. 14, but for November 2006 to February 2007 for the Northern Hemisphere.

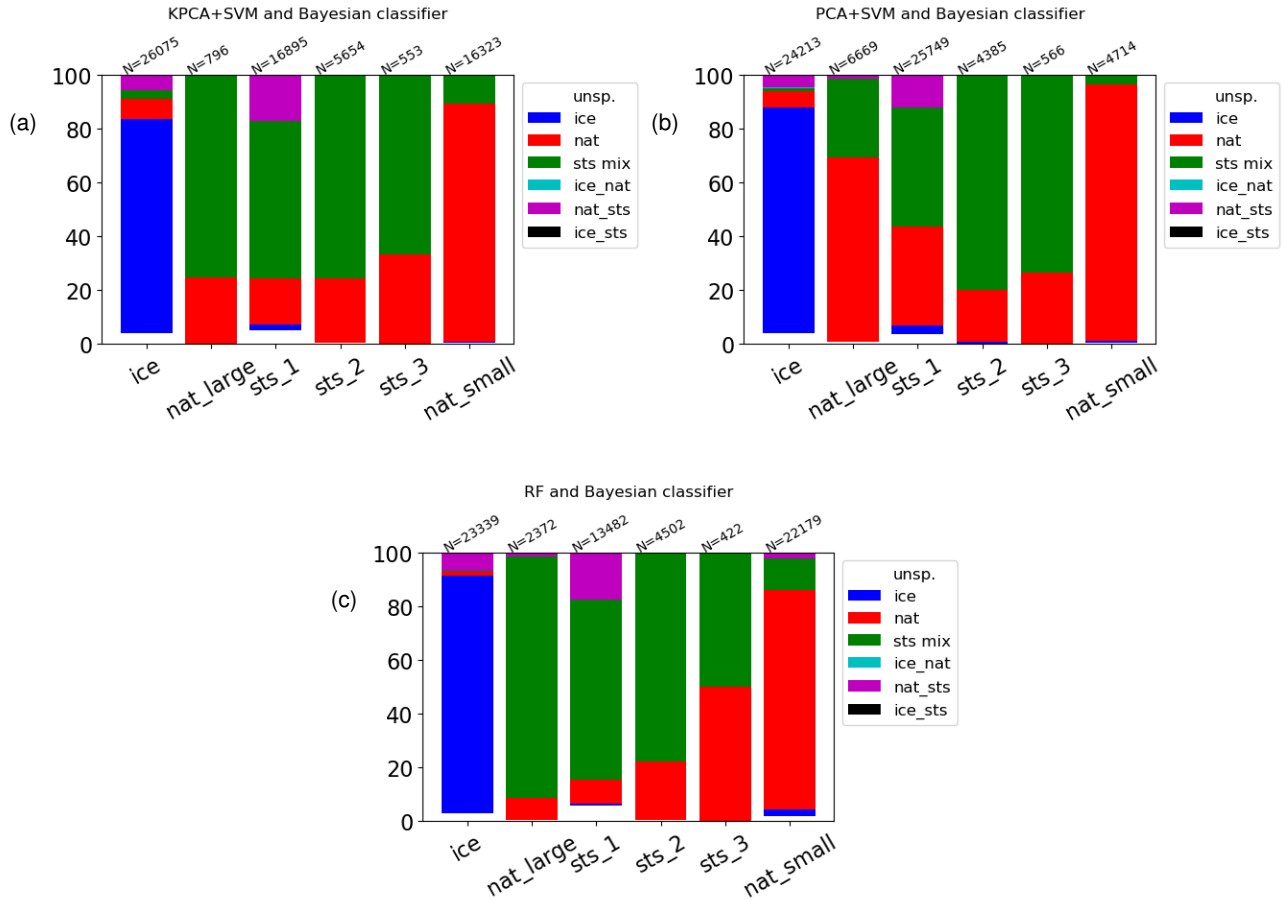

**Figure 16.** Intercomparison of ML and Bayesian classifiers for Southern Hemisphere winter (May to September 2009). Ticks on the x-axis ticks represent the classes of the KPCA+SVM classifier (a), the PCA+SVM classifier (b), and the RF classifier (c). The y-axis indicates the fraction of the classes as predicted by the Bayesian classifier. N is the number of samples belonging to each class of the ML classifiers.