# Peer review of "Exploration of machine learning methods for the classification of infrared limb spectra of polar stratospheric clouds"

_Atmospheric Measurement Techniques, 2019_

## Referee Comment (RC1) · Anonymous Referee #2 · 19 Feb 2020

In their paper, Sedona et al. explore the potential of applying machine learning methods to classify PSC observations of infrared limb sounders. To test their approach they use the Envisat MIPAS data for one Antarctic winter and one Arctic winter. Different ML techniques are tested and they find that all of them are suitable to retrieve information on the composition of PSCs, but that the random forest method seems to be the most promising one.

This is a very interesting study and deserves to be published in AMT. However, I have several major and minor comments that should be considered before publication.

**General comments and questions:**

1. A discussion on the previous classical schemes based on the optical properties of PSCs (see Achtert and Tesche, 2014) in comparison to the ML methods is missing.

2. Are ML methods really better? What is the advantage? This needs to be discussed as well.

3. The number of self citations is to high. I know that this study builds on what has been done before, however, there are several occasions (see my specific comments) where other references could be used. It simply does not correct when the entire introduction is based on Spang et al. and Hoepfner et al. citations who are not only co-authors of this study but also not the only scientists working on this topic.

4. What result would you get or could you expect when other winters are considered? Do the ML methods work in the same way for all different kinds of winters (cold or warm, dynamically more or less active)?

5. Why have the two winters presented in this study been chosen?

**Specific comments:**

P2, L5-8: "......MIPAS measurements are considered to be of great importance for the study of PSCs......" First of all, I would suggest to rephrase the sentence and to write either "are quite suitable for studying PSCs " or "MIPAS measurements have been used to study PSC processes." Second here the self citations could be avoided or at least decreased. It would be enough to cite the two recent papers by Spang et al. and Höpfner et al. (thus the 2018 papers). Even better would be if you would cite here some papers where MIPAS observations have actually been used to investigate PSCs and related processes as e.g Arnone et al. (2012), Khosrawi et al. (2018), Tritscher et

AMTD
al. (2019).

P2, L9: "...ice PSCs are generally thicker than NAT and STS (Spang et al., 2016)". This is also something which is documented in the literature and where easily another citation could be picked than Spang et al. (2016).

P2, L13: Also here, there are many more adequate citations in this context available than Spang et al. (2018).

P2, L20: Also here, avoid self citations.

P2, L21ff: Add general references on the ML methods.

P2, L32: What is the motivation for picking these winters? Where these rather warm or cold winters? Where there special dynamical conditions observed during these winters?

P3, L6: Why 14.3 orbits? Usually the number of orbits are given without position after decimal point.

P3, L30-P4, 30: If you follow the approach given in Spang et al. (2016) it would be easier if you would simply state that at the begin of the section instead of reference Spang et al. (2016) after every few sentences.

P4, L4-6: Where do you get these different composition numbers from? Are they based on the MIPAS data or on literature values? This text part is really confusing and should be rephrased.

P9, L8: "It is found that ice and small NAT accuracies are higher than the ones
of STS". Where has this been found? In this study? If yes, be more clear. Otherwise, give the according references.

P9, L21: Also here other references than Spang et al. (2018) could be given here.

P11, Summary and Conclusions: Looking at the figures I would conclude that the results derived are quite different and that is hard to say which one performs best. Thus, I have bit trouble following your reasoning, that all ML methods are suitable for the classification and that the RF performs best.

**Technical corrections:**

P1, L2: enhance  $\rightarrow$  improve

P1, L9: From the both  $\rightarrow$  From both

P1, L21: repitition of "used". Please rephrase the sentence.

P6, L19: Rearrange sentence as follows: "An interesting characteristic of the RF clasifier is that it can give by calculating the Gini index (Ceriani and Verme, 2012) also a measure of the feature importance"

**References:**

Achtert, P., and M. Tesche, Assessing lidar-based classification schemes for polar stratospheric clouds based on 16 years of measurements at Esrange, Sweden, J. Geophys. Res. Atmos., 119, 1386–1405, doi:10.1002/2013JD020355, 2014.

Arnone, E., Castelli, E., Papandrea, E., Carlotti, M., and Dinelli, B. M.: Extreme ozone depletion in the 2010–2011 Arctic winter stratosphere as observed by MI-
PAS/ENVISAT using a 2-D tomographic approach, Atmos. Chem. Phys., 12, 9149–9165, https://doi.org/10.5194/acp-12-9149-2012, 2012.

Khosrawi, F., Kirner, O., Stiller, G., Höpfner, M., Santee, M. L., Kellmann, S., and Braesicke, P.: Comparison of ECHAM5/MESSy Atmospheric Chemistry (EMAC) simulations of the Arctic winter 2009/2010 and 2010/2011 with Envisat/MIPAS and Aura/MLS observations, Atmos. Chem. Phys., 18, 8873–8892, https://doi.org/10.5194/acp-18-8873-2018, 2018.

Tritscher, I., Grooß, J.-U., Spang, R., Pitts, M. C., Poole, L. R., Müller, R., and Riese, M.: Lagrangian simulation of ice particles and resulting dehydration in the polar winter stratosphere, Atmos. Chem. Phys., 19, 543–563, https://doi.org/10.5194/acp-19-543-2019, 2019.

---

## Referee Comment (RC2) · Anonymous Referee #1 · 26 Feb 2020

An innovative study is presented, applying machine learning techniques for PSC classification in limb emission infrared spectra. A SVM-based classifier is applied, using input from PCA and KPCA feature extraction from a large set of BTDs, and a RF-based classifier using BTD features without prior feature selection. The methods are compared with an established PSC classification method reported in the literature (Bayesian classifier). Performance of the new classifiers is assessed using MIPAS data from the Northern hemisphere winter 2006/2007 and the Southern hemisphere winter 2009. Potential advantages in comparison to conventional methods are discussed.

The presented study using ML approaches is timely and clearly of interest for PSC

classification. The manuscript is well organized and mostly written clearly. However, the assessment and comparison of the new ML approaches with the conventional BC method is sometimes difficult to follow. Overlap regions and potential ambiguities of the different classification methods used are not clearly defined. Benefits of the new methods should be elaborated more clearly: why should the user decide to choose one of the new methods instead of established methods that are based on physical understanding and expert knowledge? Do the new methods provide scientifically more robust results? I recommend publication after addressing the following points:

Major points

1) To assess the performance of the new ML methods, different classification schemes are used for the 'conventional' reference method (BC) and the new ML methods. A clear comparison of the different classifications schemes, their overlap regions, and potential ambiguities is missing and should be provided prior to chapters 4.3.1 and 4.3.2. It should be clearly defined what is counted as 'ice', 'sts' and 'nat' in the discussion in these chapters (e.g., is 'nat_sts' counted to both, 'nat' and 'sts'? What about 'ice_sts'? Can 'stsmix' include ice or nat, too?). Furthermore, are the categories 'ice', 'sts' and 'nat' used in the sense of composition classes such as used by Pitts et al. (2018) in the discussion and Figs. 9-11? Or does it mean that the optical properties of these constituents can be identified/are dominating? Clear definitions of used categories should be provided, and it should be differentiated between PSC types, composition classes and constituents.

2) The benefits of the new methods should be elaborated more clearly. Are the new methods really more 'objective'? For interpretation, still comparisons with conventional data are needed, and in the end an expert needs to decide which method to trust. Are the results scientifically more robust than conventional methods that are based on physical understanding? From what has been learnt here, would it be possible to set up a robust ML PSC classification without support by a conventional method and expert knowledge?

Specific comments

1.8 is feature extraction really done from both ('these') datasets? If I understood correctly, feature extraction is done using only the CSDB but not the observations.

1.12 This sentence suggests that PCA and KPCA in combination with both, RF and SVM. If I understood correctly, PCA and KPCA are done only in combination with SVM (i.e. PCA+SVM and KPCA+SVM). Cp 8.14: 'RF...without prior feature selection'

2.2 PSCs play another important role in ozone depletion by denitrification of the stratosphere; this should be mentioned

2.3 'main types' – the choice of the terms types, constituents and composition classes should be taken with care. In reality, PSCs are often mixtures. Is 'type' used here in the sense of constituents or composition classes such as used by Pitts et al. (2018)? 'main constituents' seems more appropriate here.

2.4 What defines a 'main method'? What about airborne/balloon-borne non-optical in situ observations (mass spectrometry, chemiluminescence) and remote sensing (lidar, limb)? There are many references on other methods in the literature (e.g. Voigt et al., 2000, Molleker et al., 2014, Woiwode et al., 2016, Voigt et al., 2018). Also, microwave observations where shown to be valuable to study PSCs (e.g. Lambert et al. 2012).

2.15 'from the simulated MIPAS spectra' Is 'simulated' missing? If I understood correctly, feature selection is done only from the simulated CSDB data but not from the real MIPAS spectra

2.15 What is meant by 'type'? Composition class? Or does it mean that the optical properties of these constituents can be identified/are dominating?

2.16 'first time that ML methods ... MIPAS PSC observations' This statement should be revisited. In the literature, Bayesian classifiers such as used Spang et al., 2016 are frequently termed as ML methods (e.g. https://en.wikipedia.org/wiki/Naive_Bayes_classifier, 26.2.2020). Could the work

by Spang et al. be considered as first ML application to MIPAS?

2.24f '(PCA) and (KPCA) for feature selection, followed by... (RF) and (SVM)' Is this consistent with 8.14 'RF...without prior feature selection'? If I understood correctly, PCA and KPCA is not done in RF.

3.17ff The use of 'windows' should be revisited. Does 'spectral windows of 1 cmˆ-1' mean that the data is down-sampled or smoothed to a resolution of 1 cmˆ-1 (cp Tab. 1, the 'windows' are broader than 1 cmˆ-1)? Does 'five larger windows' mean larger than 1 cmˆ-1 or larger than R1-R8? The latter does not seem to be the case. It should be differentiated between 'spectral window' (such as R1-8 and W1-5) and spectral resolution.

3.24 What kind of 'background signals' are removed?

4.8 Höpfner et el. 2006 give an upper limit of r=3 $\mu$m instead of 2 $\mu$m for small NAT particles

4.11ff This section should be revisited. Is there a difference between 'very thin', 'thin' and 'thinnest', or is it the same? PSCs are often (very) thin clouds when compared to other clouds. What is meant by 'atmospheric variability'?

4.23 Here it would be really helpful for the reader to introduce the types or composition classes identified by the BC and discuss here or later the overlap with the ML classification and potential ambiguities. Which classes are summarized as 'nat', 'sts' and 'ice' in the later comparison with the ML results? Just a suggestion: a tabular comparison might be helpful to compare the different classifications and indicate what is counted as 'nat', 'sts' and 'ice'.

6.1ff Section 3.3 gives interesting general information about the ML methods, but the link to the presented work is somehow missing to me. How are the methods used here? What kind of input data is used and what kind of output is generated? What are the critical parameters here? The used classification scheme should be mentioned or

at least a reference to the compositions in section 2.2 should be added.

7.14ff 'a closer inspection shows. . .' In Fig. 2a, most of the area look yellowish to me; it might be helpful to adjust the color-coding. Furthermore, it might be helpful to highlight R1, R2. . . directly in Fig. 2a and 2b, since it is difficult to follow the discussion, connect regions and wave numbers with indices by using Tab. 1, and then try to identify index ranges on the panel axes. R1, R2. . . might be indicated also in Fig. 4 for easier reading.

7.17 What do the 'pronounced features' in the real an imaginary refractive index mean physically? How are they related to the spectra? Why does it make sense to feed the complex refractive indices to the ML methods, while the goal is to classify measured limb spectra and not refractive indices? At least a short explanation should be provided.

7.31 'similar clusters as Fig 2a' I have difficulties in finding the similarities, since the discussion uses wave numbers and regions while the figures use 'index'. See above: it might be helpful to indicate R1. . . somehow in the panels.

8.4 'peak in imaginary part' Which peak is meant here? 'minimum in the real part' Which minimum is meant here? See comment to 7.17: how are these refractive index features related to the spectra?

8.25 Possibly I missed it: how is the prediction accuracy determined? See Figs 12ff: How can the prediction accuracy be 99% for all methods while the classification results are relatively heterogeneous?

9.13ff Here and in in the following I got somewhat confused: For BC, the PSC classes 'unspec', 'ice', 'nat', 'stsmix', 'ice_nat', 'nat_sts', 'ice_sts' are used. For the other methods, the classes 'ice', 'nat_large', 'sts_1', 'sts_2', 'sts_3' and 'nat_small' are used. In Fig. 12-14, suddenly 'sts_mix1', 'sts_mix2' and 'sts_mix3' are used (I guess 'sts_1', 'sts_2' and 'sts_3' is meant here). In the text, the types or categories 'nat', 'sts' and 'ice' are used. The used categories and classifications should be clearly defined. Overlap regions of and potential ambiguities between the different classification schemes

should be discussed (see comments to 4.23 and 2.3).

11.33 are the new approaches really 'more objective', reminding that they need to be assessed using a 'conventional' method based on a-priori knowledge and expert knowledge, and finally one needs make a choice?

12.28 Just out of curiosity: would it be possible to make a meaningful search for further PSC constituents not covered by conventional classifications, such as nitric acid dihydrate?

Technical

2.19 approaches

3.21 have been extracted

References

Voigt et al., Science, 290, 1756–1758, 2000

Molleker et al., Atmos. Chem. Phys., 14, 10785–10801, 2014

Woiwode et al., Atmos. Chem. Phys., 16, 9505–9532, 2016

Voigt et al., Atmos. Chem. Phys., 18, 15623–15641, 2018

Lambert et al., Atmos. Chem. Phys., 12, 2899–2931, 2012

---

## Author Comment (AC1) · 23 Apr 2020

**Reply to review comments**

We thank the reviewers for the time and effort spent on the manuscript and for providing helpful comments. We considered all comments and hope that the revised draft properly addresses the open issues. Please find our point-by-point replies below (colored in blue). Note that references to figures and tables in this reply refer to the AMTD version of our paper. A revised manuscript with tracked changes has been attached to this document.

**Reviewer #1**

An innovative study is presented, applying machine learning techniques for PSC classification in limb emission infrared spectra. A SVM-based classifier is applied, using input from PCA and KPCA feature extraction from a large set of BTDs, and a RF-based classifier using BTD features without prior feature selection. The methods are compared with an established PSC classification method reported in the literature (Bayesian classifier). Performance of the new classifiers is assessed using MIPAS data from the Northern hemisphere winter 2006/2007 and the Southern hemisphere winter 2009. Potential advantages in comparison to conventional methods are discussed. The presented study using ML approaches is timely and clearly of interest for PSC classification.

The manuscript is well organized and mostly written clearly. However, the assessment and comparison of the new ML approaches with the conventional BC method is sometimes difficult to follow. Overlap regions and potential ambiguities of the different classification methods used are not clearly defined. Benefits of the new methods should be elaborated more clearly: why should the user decide to choose one of the new methods instead of established methods that are based on physical understanding and expert knowledge? Do the new methods provide scientifically more robust results? I recommend publication after addressing the following points:

We gratefully thank the Reviewer for the helpful comments and suggestions. We tried to improve the comparison of the machine learning techniques with the established method (Bayesian classifier) by carefully addressing the individual comments listed below.

Major points

1) To assess the performance of the new ML methods, different classification schemes are used for the 'conventional' reference method (BC) and the new ML methods. A clear comparison of the different classifications schemes, their overlap regions, and potential ambiguities is missing and should be provided prior to chapters 4.3.1 and 4.3.2. It should be clearly defined what is counted as 'ice', 'sts' and 'nat' in the discussion in these chapters (e.g., is 'nat_sts' counted to both, 'nat' and 'sts'? What about 'ice_sts'? Can 'stsmix' include ice or nat, too?). Furthermore, are the categories 'ice', 'sts' and 'nat' used in the sense of composition classes such as used by Pitts et al. (2018) in the discussion and

Figs. 9-11? Or does it mean that the optical properties of these constituents can be identified/are dominating? Clear definitions of used categories should be provided, and it should be differentiated between PSC types, composition classes and constituents.

We agree that the term "type" should only be used for the classical/historical PSC type 1a, 1b and 2 classification, where 2 = ice, Type 1a = solid Nitric Acid hydrates, most likely NAT, and 1b liquid supercooled ternary (H2SO4-HNO3-H2O) solution droplets (STS). This terms are not really applied in the paper. We are using composition classes for our classifiers. As an example, we can consider the BC classes ice, NAT, STS_mix where the IR spectra are dominated by a single composition (ice, small NAT, STS). The BC has also some STS_NAT may be composed of STS plus small NAT respectively. The BC has also mixed composition classes, the NAT_STS, ICE_STS and ICE_NAT. These classes are not present in the proposed ML approaches. While ICE_STS and ICE_NAT classes of the BC have a negligible population, samples belonging to the NAT_STS class of the BC, characterized by a non-negligible population, are labeled as belonging to one of the new ML classes (mostly STS 1 as it can be observed in Fig. 16 (added in the manuscript). For the proposed ML methods, we have also defined STS subclasses for the CSDB (depending on temperature the amount oh HNO3 is changing in STS) and splitted NAT into small and large NAT classes. We have added an explanation on how the classes have been defined in Sect. 3.1 an 3.3. We have also corrected the use of the terminology in the manuscript.

2) The benefits of the new methods should be elaborated more clearly. Are the new methods really more 'objective'? For interpretation, still comparisons with conventional data are needed, and in the end an expert needs to decide which method to trust. Are the results scientifically more robust than conventional methods that are based on physical understanding? From what has been learnt here, would it be possible to set up a robust ML PSC classification without support by a conventional method and expert knowledge?

A general advantage of using ML methods is that they do not require a-priori expert's knowledge. ML methods can automatically learn the prominent features from data, reversing the task of finding complex patterns from an expert to an automatic algorithm. They are objective in the premises, i.e., the training and prediction can be performed without expert's knowledge. However, it is true that evaluation in this particular study still requires the presence of an expert, due to the fact that suitable ground truth data are missing. For this reason, the only meaningful assessment we could do was to compare the classification schemes against each other. Another advantage is that the ML models can be enhanced when a new synthetic data set becomes available, making it straightforward to expand the scope of the model to new classification tasks (f.e., considering different particle types or size distributions). Moreover, ML methods can be trained quickly and inference for large data sets can be performed in a short time. A specific advantage of the proposed ML methods is that they can predict not only small NAT but also large NAT. We have tested this capability against the BC on a subset of the CSDB dataset, selecting only spectra of large NAT. While the BC cannot correctly classify them, the proposed ML schemes show promising results. We added Tab. 7 and commented on that in Sect. 4.2.

We revised also Sect. 5 to highlight the advantages of using ML methods.

Specific comments

1.8 is feature extraction really done from both ('these') datasets? If I understood correctly, feature extraction is done using only the CSDB but not the observations.

The feature extraction models (PCA and KPCA) are fitted on the CSDB only, but features are extracted from both datasets for training and prediction. We provided additional information in Sect. 3.3 to clarify.

1.12 This sentence suggests that PCA and KPCA in combination with both, RF and SVM. If I understood correctly, PCA and KPCA are done only in combination with SVM (i.e. PCA+SVM and KPCA+SVM). Cp 8.14: 'RF ... without prior feature selection'

Correct, we rephrased this.

2.2 PSCs play another important role in ozone depletion by denitrification of the stratosphere; this should be mentioned

We rephrased the sentence and added corresponding references.

2.3 'main types' – the choice of the terms types, constituents and composition classes should be taken with care. In reality, PSCs are often mixtures. Is 'type' used here in the sense of constituents or composition classes such as used by Pitts et al. (2018)? 'main constituents' seems more appropriate here.

We tried to improve the terminology (i.e., 'types', 'constituents', or 'composition classes') used throughout the paper and corrected this as suggested.

2.4 What defines a 'main method'? What about airborne/balloon-borne non-optical in situ observations (mass spectrometry, chemiluminescence) and remote sensing (lidar, limb)? There are many references on other methods in the literature (e.g. Voigt et al., 2000, Molleker et al., 2014, Woiwode et al., 2016, Voigt et al., 2018). Also, microwave observations where shown to be valuable to study PSCs (e.g. Lambert et al. 2012).

We agree that the phrase 'main method' was misleading and added more information on the different observational methods and references in the introduction.

2.15 'from the simulated MIPAS spectra' Is 'simulated' missing? If I understood correctly, feature selection is done only from the simulated CSDB data but not from the real MIPAS spectra

The feature extraction methods have been fitted on the CSDB and then used to extract the features from both, CSDB (input to the classifier for training) and from the MIPAS measurements (input to the classifier for prediction). We provided additional information in Sect. 3.3, explaining the "two steps" of feature extraction, first fitting it on the CSDB and then extracting features from the CSDB and MIPAS data sets.

2.15 What is meant by 'type'? Composition class? Or does it mean that the optical

properties of these constituents can be identified/are dominating?

Fundamentally, the classification is based on the micro- and macrophysical optical properties of the PSC particles as seen in the infrared spectra. Please see reply to 2.3.

2.16 'first time that ML methods ... MIPAS PSC observations' This statement should be revisited. In the literature, Bayesian classifiers such as used Spang et al., 2016 are frequently termed as ML methods (e.g. `https://en.wikipedia.org/wiki/Naive_Bayes_classifier`, 26.2.2020). Could the work by Spang et al. be considered as first ML application to MIPAS?

We agree that the statement could be misleading and that it needs more explanation. However, we think the statement still holds as the Bayesian Classifier used by Spang et al. was tuned empirically, i.e., the process of "learning" was trivial in a sense that it was not done by a machine. We rephrased the sentence to explain that we significantly extended the use of ML methods in this work based on more advanced learning strategies.

2.24f '(PCA) and (KPCA) for feature selection, followed by ... (RF) and (SVM)' Is this consistent with 8.14 'RF ... without prior feature selection'? If I understood correctly, PCA and KPCA is not done in RF.

The reviewer is correct, thus we rephrased the sentence. Moreover, we added a new figure in Section 3.3 to better visualize the full pipeline of the classification procedures.

3.17ff The use of windows should be revisited. Does spectral windows of 1 cm$^{-1}$ mean that the data is down-sampled or smoothed to a resolution of 1 cm$^{-1}$ (cp Tab. 1, the windows are broader than 1 cm$^{-1}$)? Does five larger windows mean larger than 1 cm$^{-1}$ or larger than R1-R8? The latter does not seem to be the case. It should be differentiated between spectral windows (such as R1-8 and W1-5) and spectral resolution.

In this study, the high-resolution MIPAS data have generally been down-sampled or smoothed to a resolution of 1 cm$^{-1}$. The reason why we did this is that such a resolution is largely sufficient to detect the broader scale spectral features used to discriminate between different PSC types. The only exception are five larger windows being broader than 1 cm$^{-1}$, which have been selected for consistency with previous work (e.g., by following the definitions of the cloud index or the NAT index). We revised the text accordingly to clarify.

3.24 What kind of background signals are removed?

Background signals arise from interfering species or instrument effects such as radiometric calibration errors. We revised the text in the manuscript.

4.8 Höpfner et el. 2006 give an upper limit of $r = 3\,\mu m$ instead of $2\,\mu$m for small NAT particles

In our paper we considered to be small NAT those NAT particles with radius up to $r = 2\,\mu m$, so actually particles with $r = 3\,\mu m$ are considered as large NAT. In Hoepfner et al. 2006 (Fig.9) it can be seen that NAT with $r = 3\,\mu m$ are not clearly distinguishable

from particles of other composition considering the 820 cm$^{-1}$ feature. In fact, the BC was already starting to misclassify spectra of NAT with $r = 2\,\mu m$. Restricting the analysis on CSDB to NAT spectra with $r = 2\,\mu m$s, the BC classifies them as NAT: 0.868590 and STS: 0.131410.

4.11ff This section should be revisited. Is there a difference between very thin, thin and thinnest, or is it the same? PSCs are often (very) thin clouds when compared to other clouds. What is meant by atmospheric variability?

We rephrased the paragraph to be more precise on how we selected the threshold. The revised paragraph now reads: "To prepare both, the real MIPAS and the CSDB data for PSC classification, we applied the cloud index (CI) method of Spang et al. (2004) with a threshold of 4.5 to filter out clear air spectra. In optimal conditions a CI¡6 detects clouds with extinction coefficients down to about 2e-5km-1 in the mid-IR (semhbi2012). However, in the polar winter regions these optimal conditions do not persist over an entire winter season. Hence, we selected a threshold of 4.5 that reliably discriminates clear air from cloudy air in the southern and northern hemisphere polar winter regions as it is sensitive to extinctions down to 5e-4km-1 (Griessbach2020, Fig. 2c)."

4.23 Here it would be really helpful for the reader to introduce the types or composition classes identified by the BC and discuss here or later the overlap with the ML classification and potential ambiguities. Which classes are summarized as nat, sts and ice in the later comparison with the ML results? Just a suggestion: a tabular comparison might be helpful to compare the different classifications and indicate what is counted as 'nat', 'sts' and 'ice'.

As suggested by the reviewer, we explained how the Bayesian Classifier assigns the output labels to the composition classes. We also added more information on the output classes of the new ML methods.

6.1ff Section 3.3 gives interesting general information about the ML methods, but the link to the presented work is somehow missing to me. How are the methods used here? What kind of input data is used and what kind of output is generated? What are the critical parameters here? The used classification scheme should be mentioned or at least a reference to the compositions in section 2.2 should be added.

We revised Sect. 3.3 and added a new paragraph linking the general description of the ML methods to the specific application in our paper. A new figure showing the flowchart of classification has been added to demonstrate the whole pipeline from the input data to the output labels.

7.14ff a closer inspection shows ... In Fig. 2a, most of the area look yellowish to me it might be helpful to adjust the color-coding. Furthermore, it might be helpful to highlight R1, R2 ... directly in Fig. 2a and 2b, since it is difficult to follow the discussion, connect regions and wave numbers with indices by using Tab. 1, and then try to identify index ranges on the panel axes. R1, R2 ... might be indicated also in Fig. 4 for easier reading.

In this revision, we restricted the color scale to remove the yellowish hue in Fig. 2a.

We also added labels to identify the regions R1, R2, ... in Figs. 2a and 2b.

7.17 What do the 'pronounced features' in the real an imaginary refractive index mean physically? How are they related to the spectra? Why does it make sense to feed the complex refractive indices to the ML methods, while the goal is to classify measured limb spectra and not refractive indices? At least a short explanation should be provided.

Physically, the real part of the refractive index characterizes the scattering whereas the imaginary part characterizes the absorption of radiance. The real and imaginary part describe the "microphysical properties" of the different PSC particles, so we would expect that the classification methods are sensitive to it (as discussed in the paper). The classification methods are using measured or simulated radiances, only, but the spectra themselves are affected by the refractive indices. We edited the manuscript to clarify on this point.

7.31 'similar clusters as Fig 2a' I have difficulties in finding the similarities, since the discussion uses wave numbers and regions while the figures use 'index'. See above: it might be helpful to indicate R1 ... somehow in the panels.

Please see reply to 7.14ff.

8.4 'peak in imaginary part' Which peak is meant here? 'minimum in the real part' Which minimum is meant here? See comment to 7.17: how are these refractive index features related to the spectra?

We added a reference to the region and discussed it in more detail. Please see reply to 7.17.

8.25 Possibly I missed it: how is the prediction accuracy determined? See Figs 12ff: How can the prediction accuracy be 99% for all methods while the classification results are relatively heterogeneous?

The prediction accuracies reported in Tab. 6 and Fig. 5 are computed on the CSDB synthetic dataset. Even though the classifiers score similarly on the synthetic data set, they may learn different mapping functions, so when deployed on real measurements the results can vary. We discussed this later in the manuscript (P12 L17).

9.13ff Here and in in the following I got somewhat confused: For BC, the PSC classes 'unspec', 'ice', 'nat', 'stsmix', 'ice_nat', 'nat_sts', 'ice_sts' are used. For the other methods, the classes 'ice', 'nat_large', 'sts_1', 'sts_2', 'sts_3' and 'nat_small' are used. In Fig. 12-14, suddenly 'sts_mix1', 'sts_mix2' and 'sts_mix3' are used (I guess 'sts_1', 'sts_2' and 'sts_3' is meant here). In the text, the types or categories 'nat', 'sts' and 'ice' are used. The used categories and classifications should be clearly defined. Overlap regions of and potential ambiguities between the different classification schemes should be discussed (see comments to 4.23 and 2.3).

We checked and corrected the class names of the ML methods used here to be consistent throughout the manuscript. We also provided more information and explanation in Sect. 3.1 and Sect. 3.3 as requested.

11.33 are the new approaches really 'more objective', reminding that they need to be assessed using a 'conventional' method based on a-priori knowledge and expert knowledge, and finally one needs make a choice?

As discussed earlier, it is true that in this study we still need expert knowledge for the evaluation. This is largely caused by the fact that no suitable ground truth data are available for the use case. However, the ML methods themselves learn complex patterns automatically from the data, without the need of hand tuning of the parameters. The ML methods are more "objective" in that they are not making assumptions on the distribution of data but rather learn from them. Thus, we reformulated our statement in Sect. 5, explaining in what sense we think ML methods are more "objective".

12.28 Just out of curiosity: would it be possible to make a meaningful search for further PSC constituents not covered by conventional classifications, such as nitric acid dihydrate?

In principle, if simulated radiance data for NAD can be generated, it would be possible to train a ML classifier for this additional task. However, it would be tricky to assess the performance of the classifier, unless ground truth data are available.

Technical

2.19 approaches

Corrected.

3.21 have been extracted

Corrected.

P11, Summary and Conclusions: Looking at the figures I would conclude that the results derived are quite different and that is hard to say which one performs best. Thus, I have bit trouble following your reasoning, that all ML methods are suitable for the classification and that the RF performs best.

Unfortunately, there is no suitable ground truth data available for validation, meaning that we can only compare the output of the classifiers against each other and provide a qualitative assessment and ranking of their performance. Overall, we think that the RF results might be most realistic, because the Bayesian classifier is known to find less NAT for MIPAS compared to CALIOP satellite observations, especially for Northern Hemisphere

winter conditions. Moreover, the RF provides a direct view on which features it uses to discriminate between the classes. In this way it becomes possible to evaluate the consistency of the classifier with respect to physical knowledge. In Sect. 5, we provided additional information on the reason why we think the RF method is the most promising one.

Technical corrections:

P1, L2: enhance → improve

P1, L9: From the both → From both

P1, L21: repitition of "used". Please rephrase the sentence.

P6, L19: Rearrange sentence as follows: "An interesting characteristic of the RF clasifier is that it can give by calculating the Gini index (Ceriani and Verme, 2012) also a measure of the feature importance"

We fixed all technical corrections.

We carefully considered the suggested references and added them in the manuscript.

[Figure]

(a)

(b)

(c)

(d)

Figure 1: Area covered by ice PSC (a) and NAT (b) in NH 2006/2007, by ice PSC (c) and NAT (d) in SH 2009.

[revised manuscript text omitted]

---

## Author Response (AR2)

**Reply to review comments**

We thank the reviewers for the time and effort spent on the manuscript and for providing helpful comments. We considered all comments and hope that the revised draft properly addresses the open issues. Please find our point-by-point replies below (colored in blue). Note that references to figures and tables in this reply refer to the AMTD version of our paper. A revised manuscript with tracked changes has been attached to this document.

**Reviewer #1**

The authors have clearly adressed the revisions and improved the manuscript. The following points should be clarified prior to publication:

1.19 sentence should be revisited: denitrification and heterogenous activation both affect ozone loss but are different things

We rephrased the sentence, which now reads: " PSC are known to play an important role in ozone depletion caused by denitrification of the stratosphere and heterogeneous reactions under cold conditions (Solomon, 1999; Toon et al., 1986). The presence of man-made chlorofluorocarbons (CFCs) in the stratosphere, which have been used for example in industrial compounds present in refrigerants, solvents, blowing agents for plastic foam affects ozone depletion."

2.19 Voigt et al. (2018) use airborne remote sensing, which does not fall in the categories in situ or microwave

Removed misplaced citation.

8.13ff More explanation is required for Fig 2. Should the reader focus on rows/columns, or the diagonal? or maximum values only? What about the "W" features? They are missing in the discussion and labels

Fig. 4 (formerly Fig. 2) has been updated with the suggested technical corrections. The reader should focus on where the maximum values are located (i.e. which are the features with largest variance in Fig. 4a and most important features in Fig. 4b and the indices of the BTs from which they were computed). The explanation of Fig. 4 has been expanded, considering the "W" features as well: "Considering the larger windows W, the matrices of the variance and of the RF feature importance seem to agree, with the exception of W3 ($\sim 820$ cm$^{-1}$) that is regarded as important by the RF scheme but is not characterized by high variance, confirming the capability of the RF of detecting the NAT feature."

8.14f 820-840 cm$^{-1}$ region is discussed already in previous sentence; what about the very strong feature at BT index ca. 10?

The feature at BT index ca. 10 was already mentioned in the manuscript but we added

the BT index to make it easier to identify it.

9.2 R3 seems relatively important and R5 unimportant, both opposite to Fig. 2a. Should this be discussed? Could this explain the better performance of RF?

We think that this is something to be explored. We have observed that there are a some regions where the RF selects important feature even if the variance is low. To better clarify on this point, we rephrased a portion of the paragraph, which now reads: "BTDs between 1224 to 1250 cm$^{-1}$ (R3) and 1404 to 1412 cm$^{-1}$ (R4) are also regarded as important. The RF features located in this cluster is in contrast with the relatively low BTDs variance in the same area. A similar observation can be done regarding BTDs between 782 to 800 cm$^{-1}$ and 810 to 820 cm$^{-1}$ (both belonging to R1). This region is in the range of values of the NAT feature, providing a possible explanation of the ability of the RF to detect the characteristic peak of small NAT as well as a shift with the increase of the radius."

9.10 BT indices should be provided for the 790 and 1235 cm$^{-1}$ features to allow locating these features in Fig. 2

We added the BT indices for those features. The feature located at 790 cm$^{-1}$ (index ca. 10) was also mentioned in previous point 8.14.

11.10 In the seasonal analysis, only the classes sts, ice and nat are shown. Does this mean that also mixtures containing sts are summarized as "sts" etc.? Is "nat_sts" counted as both "nat" and "sts", thus both in Fig 10a and 12a?

The nat_sts class of the Bayesian classifier is not considered in Fig. 11a and 13a (formerly Fig. 10a and 12a). We decided to consider in the seasonal analysis only the single composition classes of the Bayesian classifier. This is because the nat_sts class could be misleading. In fact, looking at Fig. 2, it can be seen that nat_sts is predicted by the new ML methods mostly as sts or ice and only a small fraction as nat (small or large). We added a disclaimer at the beginning of the seasonal analyses section, which reads: " The mixed composition classes of the Bayesian classifier (NAT_STS, ICE_STS and ICE_NAT) are not considered in this analysis."

11.30 A recent study indicates that large aspherical "beta-NAT" particles can be discriminated in ir spectra (Woiwode et al., 2019). Are the methods used here sensitive to such particles?

For the CSDB only spherical particles were considered. Hence, we cannot say anything about strongly aspherical particles as discussed by Woiwode et al. 2019. In case a database including simulations of large aspherical particles becomes available, it would be possible to extend the analysis to them. At the moment, the CSDB includes large spherical NAT particles, that were not detected by the Bayesan classifyer, but by the ML methods, as shown and discussed in Sect. 4.2 of the manuscript.

12.11ff Which part of the subsequent section belongs to Fig 15. and which to Fig. 16? Is Fig. 16 discussed at all? The discussion is short for 2 figures including 6 panels in total

We revised the section, making clear reference to the figures during discussion and elaborating more on both of them.

12.13 "almost all" seems too strong

Inserted percentages in the text.

12.16 Do percentages in text refer to y-axes in Fig. 15? "nat_small" seems red > 80%, while the text says "around 50 % and 60 %". Some explanation is required on how percentages given in the text are related to the figures

This paragraph has been corrected. It now reads: " Concerning NAT, the RF classifier predicts as small NAT more than 80% of what had been classified as NAT class by the Bayesian classifier (Fig. 2). The PCA+SVM and KPCA+SVM methods predict a smaller fraction of small NAT for the NAT class of the Bayesian classifier, around 30% and 70%, respectively. The PCA+SVM in particular is predicting a significant smaller amount of samples belonging to the small NAT class than the other methods (Fig. 16), while it predicts a larger number of samples of the STS subclasses. This result may suggest that PCA+SVM and KPCA+SVM are not as sensitive as BC for small NAT detection, while RF is."

Technical

3.1 "((" and "))"

7.31 numbers of figures should be ordered according to appearance in text; "pipeline" Fig 4 = Fig 2?

15.6 2010-2011 Fig 2 suggest to move "R" labels to middle of respective windows; otherwise one may get the impression that R1 etc are lines only; increments of y and x-axis should be identical Fig 4 2x"radianCES"; explain "f.e." Fig 5 grey/dark grey shading should be labelled or defined in caption Fig 15,16 define N

We took into account all the suggested technical corrections and updated the figures.

P2, L4: Rephrase this sentence as well. There are three types of PSCs. At least that is what you are meaning here.

In our first draft we referred to them as "three main types", but following reviewer #1 comment in the previous revision step (minor comment 2.3) we decided to adopt the term constituent, since PSCs can also appear in mixtures.

P2, L6: observations →observation

P2, L12: with LIDAR →by lidar

P2, L13: remove parantheses around the Biele reference (citet instead of citep)

P2, L14: move "(type)" at the end of the sentence.

P3, L1-2: remove the parantheses around the references appear twice)

P3, L9: seasons →season

P3, L16: in STS →of STS

P3, L19: Weren't these termed low resolution and high resolution modes?

In (Raspollini, P., Carli, B., Carlotti, M., Ceccherini, S., Dehn, A., Dinelli, B. M., Dudhia, A., Flaud, J.-M., López-Puertas, M., Niro, F., Remedios, J. J., Ridolfi, M., Sembhi, H., Sgheri, L., and von Clarmann, T.: Ten years of MIPAS measurements with ESA Level 2 processor V6 – Part 1: Retrieval algorithm and diagnostics of the products, Atmos. Meas. Tech., 6, 2419–2439, https://doi.org/10.5194/amt-6-2419-2013, 2013) they are referred to as full and optimized resolution modes.

P4, L22 and 23: Make a proper <= sign with latex

P4, L25ff: use superscript for the exponential numbers given in the text.

P10, L27: Abbreviation SH has not introduced and only used here, so better to write also here Southern Hemisphere instead of SH

P11, L11: times series →time series

The technical corrections have been made.

[revised manuscript text omitted]